# EIS Investigation of the Corrosion Behavior of Steel Bars Embedded into Modified Concretes with Eggshell Contents

Yuri Alexandre Meyer [1], Izabella Menezes [1], Rodrigo Silva Bonatti [1], Ausdinir Danilo Bortolozo [1,2] and Wislei Riuper Osório [1,2,*]

1   School of Technology, Department of Material Science Engineering, Campus I,
    University of Campinas/UNICAMP, Limeira 13484-332, SP, Brazil; meyeryuri@gmail.com (Y.A.M.);
    zazaa_menezes@hotmail.com (I.M.); robonatti@yahoo.com.br (R.S.B.); ausdinir@unicamp.br (A.D.B.)
2   School of Applied Sciences, FCA, Research Group in Manufacturing Advanced Materials (CPMMA),
    Campus II, University of Campinas—UNICAMP, Limeira 3484-350, SP, Brazil
*   Correspondence: wislei1@unicamp.br

**Abstract:** This investigation is focused on evaluation of the corrosion behavior of embedded steel bars (SB) into concretes. Conventional and modified concretes with eggshell are prepared. Although the effect of calcium carbonate on mechanical behavior is recognized and reported, their effects as eggshell (ES) particles replacing portions of sand and cement contents are reasonably scarce. Corrosion behavior is evaluated by electrochemical impedance spectroscopy (EIS) and the potentiodynamic polarization technique. Equivalent circuit and porous electrode behavior are also considered. The novelty concerns a promising use of concrete with ES content to maintain corrosion resistance concatenated with reasonable structural properties. For this purpose, three distinct concrete mixtures are proposed, i.e., a reference and two modified concretes. One replaces 10 wt.% with cement and another 10 wt.% with sand content. It is found that porous electrode behavior helps to predict the corrosion mechanism. Finer ES particles in concrete mixture provides a rapidly passivation on rebar. This reflects positively in corrosion current density after long-term immersion. Additionally, an environmentally friendly aspect associated with economical factor constitutes a promise use of the concrete.

**Keywords:** eggshell particles; cement consumption; corrosion resistance; compressive strength; lightweight effect





## 1. Introduction

Both aviculture and aquaculture demonstrated growth in recent years [1–3]. Associated with this, an ecological pressure to dispose millions of tons of shell wastes and other skeletal wastes is provoked [3]. These residues are dumped in landfills or deposited along coastal regions that create environmental hazards and serious health risks [1].

In order to contribute to the disposal of these wastes and to establish a bridge between ecological and economic interests, it is essential to convert the wastes into products with some additional added value that are utilizable. Jinshan et al. [3] and Islam et al. [4] have reported that shells are calcium carbonate-rich sources. Investigations demonstrating the conversion of shells into limestone content to be used as promising building materials have been reported [5,6].

Limestone from grinding calcitic rocks obtained from quarries is used as a filler in concrete [7]. It acts in hydration cement improving early age hydration and strength cement [7]. In particular, eggshell powder has also been used in orthopedic cements [8] and also to increase the strength of natural rubber [9] and polymer-rich composites [10]. It was also reported that lime powder obtained from eggshells is an alternative material to natural lime [11]. Their utilization as filler in cement mortar is also reported [12]. Shih-Ching et al. [13] reported

eggshells in synthesized hydroxyapatite powders with biomaterial applications. There exists a great amount of utilization of eggshells in distinctive fields.

Considering civil engineering applications, Matschei et al. [14] have reported that calcium carbonate has two interesting and distinct roles in cement hydration. The first is to participate as an active agent in hydration and the other is as an inert load. Based on this, it is assumed that mechanical properties can be modified when eggshells are added or replacing with sand and cement. In the last decade, calcium carbonate processed as nano-particle $CaCO_3$ has widely been reported [15–20]. Wang et al. [21] recently reported that limestone replaced with cement improved certain properties (e.g., the workability, compressive and flexural strengths, and water permeability). Research showed that workability is improved due to limestone filling the voids of cement, and porosity is decreased [21].

Concerning the mechanical properties, when fine aggregate is used and a fine limestone as filler is used, the mechanical behavior is improved [21,22]. However, depending on the particle size, some properties can be differently affected. For instance, the workability is affected when a particle size varying from 0.7 μm to 15 μm is used. Bosiljkov [23] demonstrated that a finer $CaCO_3$ fill avoids cement particles, and the compressive strength is increased. Ghafoori et al. [24] also demonstrated that the compressive behavior is improved when finer $CaCO_3$ particles (~5 μm) are used compared with coarser particles (between 10 μm and 20 μm).

Additionally, Shaikh and Supit [25] have reported that the highest compressive strength investigated is that of the 1 wt.% of $CaCO_3$ nano-like particles (between 40 and 50 nm) when replacing cement content (ordinary Portland cement, Type I, ASTM C150). They also reported improvements in water adsorption and chloride penetration resistance (~22% higher than cement paste) [25]. They clarified that both the size and content percentages of $CaCO_3$ have important roles regarding both the fresh and hardened properties. Meddah et al. [26] verified that when cement is replaced with $CaCO_3$ particles more than 15 wt.%, the compressive strength is drastically decreased [26].

Investigations concerning to corrosion of reinforcing steel bars (rebar) in concrete with calcium carbonate particles as additional or replacing the cement are scarce. The corrosion of rebar is a major cause of civil infrastructure degradation in worldwide [21]. Wang et al. [21] demonstrated that chloride permeability decreases substantially when limestone replaces the cement and when the limestone particle size is increased. This is associated with filler effect since a finer limestone replaces cement [21]. Wang et al. [21] also stated that the corrosion resistance of concrete is affected by the electrical resistivity.

When this is increased, the movement of electrons is hindered, and the corrosion resistance is also increased. Ramezanianpour et al. [27] reported the effect of limestone powder on corrosion behavior of concrete. They demonstrated that electrical resistivity decreased with the increase of the fine limestone content. This is associated with nano-particle of limestone filling capillary pores and modifying C-S-H gels of concrete [28,29]. It was also reported that eggshell, calcite, and marble decreased the porosity of concrete around the steel surface [30,31].

For instance, $CaCO_3$ reacts with cement in the hydration process provoking certain roughness around the rebar. This blocks the way of the chloride ions to the steel surface rendering a coat with higher resistance [30,31]. Although $CaCO_3$ was considered as filler, it was also suggested that $CaCO_3$ commonly accelerates chemical reactions with tricalcium aluminate to constitute the carboaluminate compound, mainly when nano- or fine-like $CaCO_3$ particles are used [19].

Diab et al. [32] stated that the corrosion behavior of reinforced steel in concrete with limestone partially replacing cement is increased. They also stated that the compressive strength is strongly dependent of the fineness of $CaCO_3$ particles with respect to cement. Research found [32] that the corrosion resistance (by analyzing potentiodynamic polarization curves) increases with the increase of $CaCO_3$ fineness level. The water/cement ratio, and kind and content of cement are parameters to be carefully adopted and controlled [32].

Pech-Canul and Castro [33] also reported that higher water/cement ratios displace the more active potential region suggesting a decrease corrosion resistance. Due to the alkalinity of concrete (pH ~13), a passive protection is commonly prevalent [34,35]. Summarizing, it is consensually reported [14,21,27–29,32] that finer $CaCO_3$ particles replacing cement increases corrosion resistance. Essentially, due to $CaCO_3$ acts as filler agent and its participation in hydration will block the sites for chloride penetration or modifying the electrical resistivity.

Electrochemical impedance spectroscopy (EIS) has been used to predict and understand the corrosion behavior of reinforced steel concrete [36]. Due to the heterogeneous characteristic of cement, water/cement ratio, cell kit construction (kind of counter electrode), characteristics of rebars and other parameters, the interpretation of EIS results is a laborious task, and this technique still requires in-depth investigations [36,37].

Ghorbani et al. [38] recently used EIS measurements to evaluate the corrosion resistances of steel rebar in concrete containing fine (between 1 μm and 10 μm) marble/granite powder contents (CaO- and $SiO_2$-rich compounds, respectively). They concluded that the 10% and 20% (wt.%) of cement partially replacing with granite waste indicated improvements in corrosion resistances. Similar interpretations were also utilized when Sohail et al. [39] investigated the corrosion performance of distinct steel reinforcing bars with Portland (CEM type I).

They schematically depicted the physical interpretation of the interfaces constituted when a steel reinforced concrete is considered, i.e., bulk concrete with solution, concrete with steel rebar and the double layer-electrode interfaces. Three capacitive semi-arcs were proposed at different frequency ranges, i.e., at high and low frequencies, and the bulk concrete and rebar conditions (double layer and interfacial reactions) were characterized [37–39]. In some conditions, a Warburg element can be constituted due to ions diffusion throughout an oxide layer [37].

However, the equivalent circuit (EC) interpretation also constitutes a difficult to prescribe real rebar application and in some cases, there exists a great difficulty in comparing with other previous investigations. This due to the different size samples, types of counter electrodes, construction and configurations of corrosion cells, and the nature/characteristics of the examined concrete mixtures.

Some drawbacks are also constituted when laboratory measurements and field applications are not compared. Review papers [40,41] have been recently published detailing the corrosion mechanisms of rebar in both chloride and carbonated media. Some drawbacks concerning the electrical method for monitoring corrosion have also been reported (e.g., perturbation due to a counter electrode area lower than the rebar arrangement) [40,41].

Few studies report the EIS to predict the corrosion behavior of concrete containing $CaCO_3$, such as eggshell-like particles. Matschei et al. [14] reported that calcite ($CaCO_3$) reacts with cement, and two functions were reported. One showed that it participated as an active agent in the hydration and another as inert filler [14]. It has been reported that corrosion products are located not only on steel rebar's surfaces but also penetrate throughout the cement paste [40–42]. When measuring and analyzing (EDX element maps) the corrosion layer at the rebar/concrete interface [42,43], both Fe and Ca ions interacting with and constituting the corrosion layer were identified [42–44]. This indicates that complex by-products constitute the corrosion layer, mainly when $Ca^{+2}$, $OH^-$ and $Cl^-$ species are participating.

These species constitute a complex corrosion layer, and a porous electrode behavior seems to potentially be associated. Distinct equivalent circuits (EC) to predict the phenomenological reactions were proposed. Although a physical interpretation and configuration considering EC were proposed, due to the nature and characteristic of cement, sizes and morphologies of the aggregates, immersion solution, and other parameters, the empirical EIS plots are considerably different to the theoretical preposition.

Additionally, comparisons among distinct investigations are scarce. The experimental investigations conducted in the rebar corrosion with interpretation of initial phenomena

(e.g., double layer and porous electrode behavior) provided by EIS plots are also scarce or absent. The porous electrode is reasonably similar to a Warburg-like behavior as stated in Levie's theory [45]. MacDonald and collaborators [46,47] reported a study of penetration of the porous concrete by chloride describing corrosion behavior.

Bastidas [48] also reported Warburg impedance elements in order to provide interpretation of impedance with porous electrode behavior and diffusional process. This phenomenon will be discussed and detailed. Both the cylindrical and irregularly-shaped non cylindrical pores were modeled and investigated by Bastidas [48]. This study provides a systematic report concern to the porous electrode modeling using distinctive and previously methods reported (e.g., De Levie theory) [48].

The main aim of this proposed investigation concerns the evaluation of both the mechanical and corrosion behavior when eggshell (ES) from chicken origin is used. With this, it is expected that the mineralogy of concrete is not substantially affected. However, the economical aspect is an interesting characteristic since no decreasing in corrosion is attained. Associated with this, the novelty of this contribution concerns the evaluation of the corrosion behavior concatenated with mechanical properties. In addition, both porous and planar behaviors are involved in predicting the resulting corrosion behavior. In this investigation, three distinct concrete mixtures are prepared, i.e., a reference (control) and two modified concretes.

When cement content is replaced with eggshell particles, a percentage of about 10 wt.% of finer $CaCO_3$ (size close to cement fineness) is utilized. On the other hand, when sand content is partially (~10 wt.%) replaced with $CaCO_3$ particles, coarser eggshell particles (between 2 and 4 mm) with a 10 wt.% are used. These selection mixes are based on studies when the compressive strengths in concretes with eggshell contents were previously reported [14–26]. The concatenated mechanical behavior and corrosion resistance associated with lightweight effect are also examined.

## 2. Materials and Methods

### 2.1. Materials

A high early strength (HES) Portland cement was used in order to prepare all concrete mixtures. This cement has finer characteristics and higher hydration (rapid) than other cements. The specific mass and unit mass were 3.12 g/cm³ and 1.03 g/cm³ as also previously reported [49,50]. Fine and coarse aggregates were natural quartzite sand and basalt gravel #0 (size between 4.75 and 9.5 mm). This was adopted based on previous investigations [49,50] considering the cover area between particles and steel rebar embedded. The sand proportions were kept at 100 (±5) °C for 24 h before the mixture procedure.

No superplasticizer additive was used. This point constitutes a limitation of this study. Although a superplasticizer could be used as well as other water-to-cement (w/c) ratios, a w/c ratio of 0.6 was adopted. This is based on the fact that other modified concrete mixtures have not reached the minimal workability (slump) results. For instance, when a low w/c ratio (e.g., ~0.5) is used, no workability is reached. It is recognized that w/c ratios between 0.55 and 0.6 are commonly utilized [11,29].

Eggshell (ES) particle contents were used to partially replace with sand and cement portions. The ES particles ($CaCO_3$) from aviculture were cleaned, dried (100 ± 5 °C for 24 h) and ground. Two distinct ES sizes were utilized. One ES portion was designated as coarser ES (CES) particles, which were ranging between 2 and 4 mm. These ES particles were partially replacing 10 wt.% of the sand content. Another portion was designated as the finer eggshell portion, which replaced 10 wt.% of the cement content. These replacements were adopted based on the magnitude of the particle sizes. The CES particles were used to obtain those fine ES particles. For this purpose, an agate mortar/pestle (ceramic material) was utilized.

A mechanical grinding (~5 min.) inside a mortar/pestle assemble was conducted [48]. These sizes and portions were adopted based on previous studies when $CaCO_3$ particles replace the cement content; and mechanical strength is commonly increased [14,21,23–26].

Research has consensually reported [14,21,27–29,32] that finer $CaCO_3$ particles replacing cement, the resulting corrosion resistance is increased as previously mentioned. It was also reported that mechanical behavior is improved with $CaCO_3$ particle inclusion. However, consideration of these particle effects upon the corrosion behavior evaluated using EIS technique is absent.

The steel bars (SB) or rebars used to be embedded into the distinct concrete mixtures are chemically similar to SAE 1020 steel (composition, in wt.%, C is 0.18–0.25, Mn is 0.3–0.5, Cr and Ni are lower than 0.01%, Cu and Mo are lower than 0.2 and 0.01, respectively, with Fe in balance). These are hot-rolled, and a mill-scale is commonly formed in the surface of the final reinforcing steel bar [40–42,51,52]. Steel bars (rebars) have non-uniform microstructures and microconstituents. A mill-scale (sizing between 50 and 100 μm), a tempered martensite layer (~500 μm) and a ferrite/pearlite constitute the composition of the rebar. These layers depend on the rebar diameters and the manufacturers considered [40–42,51].

### 2.2. Concrete Mixtures

The constituents of concretes are parameterized, such as a 0.6 w/c ratio, coarse aggregate (gravel #0) and steel bar. Table 1 shows the mix proportions used in each one of the proposed concrete mixtures. The samples were designated as CC (conventional concrete), and SAND was the concrete sample with 10 wt.% sand is replaced with coarser ES particles. Finally, a sample designated as the CEM sample was that of with 10 wt.% cement content being replaced with finer ES particles. Triplicate molded concrete specimens for the compressive strength determination were considered. Similarly, triplicates were used to produce specimens to determine the tensile strength using an indirect relation with cylindrical samples [53,54]. Duplicates were considered for EIS and potentiodynamic polarization test measurements.

**Table 1.** The concrete mix proportions (CC = 1:2.4:2.57) using distinct ES particle contents replacing the sand and cement portions.

| Mix | Cement (kg/m$^3$) | Sand (kg/m$^3$) | CA (kg/m$^3$) | Water (kg/m$^3$) | ES Content (kg) | w/c Ratio | ES Average Size (μm) |
|------|------|------|------|------|------|------|------|
| CC | 368 | 884 | 946 | 221 | - - - | 0.6 | - - - |
| SAND | 370 | 799 | 950 | 222 | 80 (10%) | 0.6 | Between 2000 and 4000 |
| CEM | 339 | 903 | 967 | 204 | 34 (10%) | 0.6 | <20 |

### 2.3. Fresh and Hardened Properties Determinations

The fresh state properties of the distinct SCC samples were determined using a Brazilian standard ABNT NBR 15823:2017 [49,50]. According to ASTM C1611-18 [49,50], the traditional slump measurements were performed. These procedures were conducted in order to measure the ability of the concrete to pass through obstacles and to flow under certain conditions [11].

The hardened conditions of the proposed SCC samples are represented by both the compressive strengths and tensile (diametrical compression) strengths over 7 and 28 days, according to ABNT NBR 5739:2007 [55] and NBR 7222:2011 [56], respectively [55–57]. For this purpose, a loading speed of about 0.05 (±0.015) MPa/s was adopted. As also previously reported [57], the tensile strength was determined using Equation (1).

$$F_t = 2P/\pi Dh \tag{1}$$

where $F_t$ is the tensile strength (N/mm$^2$), P is the maximum load of rupture (N) and D and h are the diameter height of the sample (mm) used.

### 2.4. EIS and Potentiodynamic Polarization Measurements

In order to perform the electrochemical measurements, a conventional three-electrode cell kit was used. The embedded steel bars (diameter ~5.5 ($\pm$0.2) mm) constitute the working electrodes (WE) with a total area of about 900 ($\pm$20) mm$^2$. A conventional Ag/AgCl, SCE (saturated calomel electrode) as the reference electrode was adopted. Four graphite bars (of about 4500 $\pm$ 100 mm$^2$) as counter electrodes were selected. These are displaced at 120$^\circ$ around the concrete sample. This constitutes an experimental arrangement similar to those previously reported [38,40,58].

Figure 1a depicts the experimental arrangement of the embedded steel bar for the electrochemical corrosion behavior of the proposed concretes.

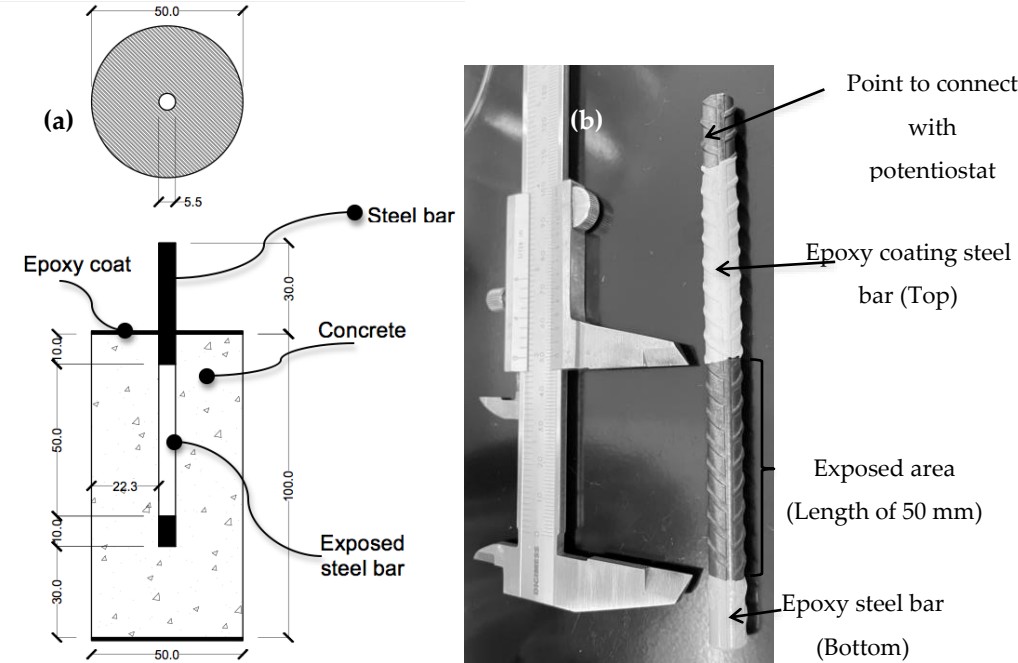

**Figure 1.** (**a**) Experimental representation of the embedded steel bar for electrochemical corrosion behavior in stagnant and naturally aerated 0.5 M NaCl at 25 ($\pm$3) $^\circ$C. Dimensions are in millimeters. (**b**) A typical steel bar coated with epoxy before embedding into the concrete used in experiments.

Figure 1b shows a typical steel bar with epoxy coating at bottom and top of the steel bars before being embedded into concrete. A similar arrangement was previously reported [57]. Before all experiments, the samples were immersed in stagnant and naturally aerated 0.5 M NaCl solution for 7 days. The samples were displaced in recipient containing NaCl with concrete bulk completely immersed (i.e., of about 100 mm considering the height of the cylindrical concrete sample.

The top of the steel bar (i.e., ~30 mm) was not immersed in NaCl solution. After this period (stationary stage), in order to procedure both EIS and polarization tests, a volume of about 450 ($\pm$15) mL of a stagnant and naturally aerated 0.5 M NaCl at 25 ($\pm$2) $^\circ$C with an initial pH of about 6.5 ($\pm$0.5) was adopted. For each new experiment, the electrolyte was replaced. The concrete specimens prepared for electrochemical tests were kept immersed in NaCl solution at room temperature during all curing periods.

Before initiating all electrochemical measurements, the concrete specimens were kept immersed in NaCl solution for 10 min with all electrodes connected. EIS measurements were conducted after about 15 min, due to a quasi steady state being attained. This is commonly practiced intending to standard all measures and to stabilize distortions and oscillations affecting the resulting measurements [38,48,59–61]. EIS experiments were conducted (~40 min) before the potentiodynamic polarization techniques were conducted (~1 h).

A VersaStat 4, Princeton Applied Research® (PAR) supplied by "Interprise Analytical Instruments" (Paulínia, SP, Brazil) was used. EIS tests were conducted using 10 points per decade with potential amplitude ~10 mV, peak-to-peak (AC signal) in open-circuit. A frequency range between $10^5$ and $10^{-2}$ Hz was adopted as previously reported [59–63]. Complex non-linear least squares (CNLS) simulations were performed and compared with the experimented impedance data. For this purpose, a ZView® software (version 2.1b) and Microcal Origin® were used. Considering the potentiodynamic polarization measurements, these were performed after EIS measurements using the same potentiostat described.

A scan rate of 0.167 mV/s scanning from −200 to −900 mV was adopted. The corrosion current densities ($i_{corr}$) were determined by using the concept of Tafel's extrapolation method. Both the cathodic and anodic branches were considered. The averages of corrosion potential ($E_{corr}$) and $i_{corr}$ were considered, and at least duplicates were considered. Potentiodynamic polarization measurements did not substantially affect or damage the steel bars (SB). Although it is recognized that EIS is non-destructive testing also no substantial damages on steel bar surfaces (covered by concrete bulk) were observed when polarization tests were performed. Although the top SB (portion out of the bulk concrete) depicts a red rust portion, this was cleaned (grounding) before each one of the experiments. This is made to connect with potentiostat by using a jack connection as shown in Figure 1b. After ~365 days, no substantial damages were observed.

## 3. Results and Discussion

### 3.1. Hardened and Fresh States

Figure 2a,b show the obtained ES particles utilized to the sand replaced with coarse ES particles. Figure 2c depicts the fine ES particles that replace cement content. Figure 2d,e demonstrate typical samples subjected to compressive and tensile (diametrical compression) tests, respectively. A typical fractured sample with ES replacing sand content (after compressive testing) is shown in Figure 2f. Considering hardened and fresh properties, these states were evaluated in order to verify the minimal conditions of workability and mechanical properties of the structural concretes proposed. Table 2 shows the experimental results of the slump and the compressive strength (CS) and tensile strength (TS) after 7 and 28 days of curing.

**Table 2.** Experimental results of slump, compressive and tensile (cylindrical compression) strengths at 7 and 28 days for the three distinctive concrete specimens examined.

| Mixture | CS at 7 Days (MPa) | CS at 28 Days (MPa) | TS at 7 Days (MPa) | TS at 28 Days (MPa) | Slump (mm) |
|---------|--------------------|--------------------|--------------------|---------------------|------------|
| CC      | 18 (±2)            | 23 (±2)            | 2.5 (±0.3)         | 3.1 (±0.3)          | 88 (±3)    |
| SAND    | 14 (±2)            | 15 (±2)            | 2.4 (±0.2)         | 2.8 (±0.4)          | 76 (±4)    |
| CEM     | 16 (±2)            | 18 (±2)            | 2.7 (±0.3)         | 4.9 (±0.3)          | 77 (±3)    |

The slump results suggest that all examined concretes reached their workability levels (i.e., higher than 70 mm). A 0.6 w/c ratio was considered. These results are similar to those previously published when conventional and modified concretes were examined [57,64–66].

Considering the CS and TS results over 7 and 28 days, we confirmed that the proposed concretes with sand and cement contents replaced with 10 wt.% ES particles reached the minimal structural application.

The experimental correlations between the tensile (TS) and compressive strengths (CS) at 7 and 28 days of curing are shown in Figure 3. Although distinct modified concretes and cementitious materials were reported in previous studies [50,65,66], the obtained equations describing TS = k $CS^{0.5}$ are similar to those previously reported. When the ES portions (10 wt.%) replace cement and sand contents, slight decreasing trends were provided when compared with the CC sample. This is more noticeable in the CS results. Since the mechanical behavior and fresh states of the proposed concretes are acceptable,

the corrosion resistances by using EIS and polarization curves were analyzed as discussed in the next section.

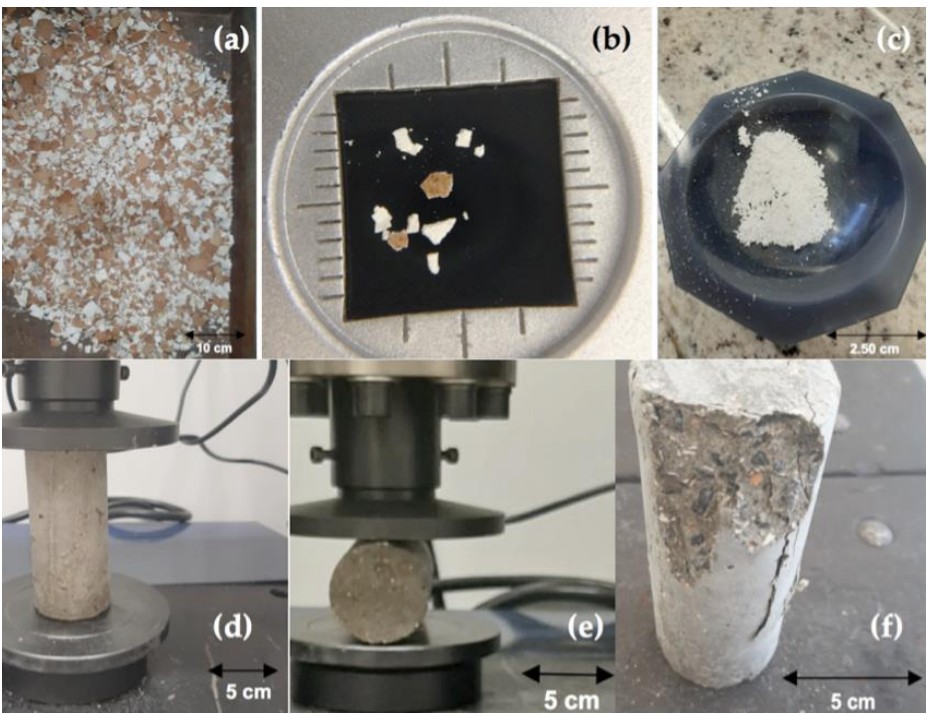

**Figure 2.** (**a**) The obtained eggshell (ES) portion, and (**b**) stereoscopic image showing the dimensions of coarser ES particles used to replaced with sand content, (**c**) finer ES particles inside agate mortar, (**d**,**e**) typical specimens subjected to compression and diametrical compression (indirect tensile behavior) and (**f**) typical fractured sample with sand content replaced with the coarse ES portion.

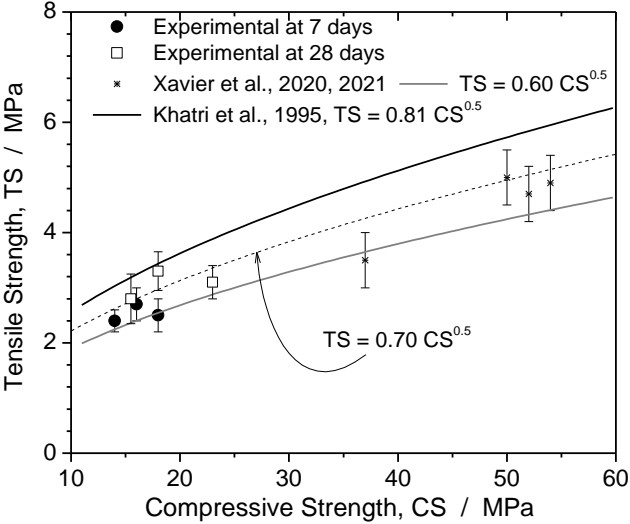

**Figure 3.** Correlation between tensile and compressive strengths demonstrating TS = k $CS^{0.5}$, considering 7 and 28 days of curing.

### 3.2. EIS Analyses: Numbers of Time Constants

In order to analyze the corrosion behavior of the examined samples, firstly, it is intrinsically demanded to determine the numbers of time constants to correlate with each reaction occurring. In order to understand the corrosion mechanism, it is important to recognize the numbers of the time constants prevalent in the system examined. Firstly, this helps to

determine the reactions occurring in an electrochemical system and to select an adequate equivalent circuit to simulate the experimental results and to evaluate the impedance parameters. These contribute to a quantitative analyses and permit the participation of each element in the corrosion mechanism. This task also constitutes a novelty and will be useful for the future researchers.

Considering EIS plots, there are distinctive regions defining the electrochemical behavior of the examined concretes [32,36–38,59–62]. At $10^5$ Hz and ~$10^3$ Hz is the high frequency (HF) describing the electrolyte resistance in the bulk of the concrete [32,36–38]. Between frequencies ~$10^3$ Hz and ~$10^0$ Hz, the double layer formation is represented. A third region is characterized at lower frequency (LF), commonly lower than $10^0$ Hz (up to $10^{-2}$ Hz), which describes the interface reactions [36–38,59,60] between the surface of the steel bar with the electrolyte penetrated into concrete (paste) with its characteristics (porous, aggregates and intermediates ions).

Considering Bode diagrams, it is briefly stated that these diagrams are a semi-logarithmic coordinate plot of the transfer function of a linear time-invariant system versus frequency. The frequency at the horizontal axis is a logarithmic scale, and the frequency response of the system is depicted using a Bode plot. Additionally, it is generally composed of two diagrams, one amplitude-frequency diagram represents the change of the decibel value of the frequency response gain against frequency, and another phase-frequency diagram is the change of the phase of the frequency response against the frequency.

When Nyquist plots are considered, these are for continuous-time linear time-invariant system; the gain and phase of its frequency response are plotted in the complex plane in polar coordinates. These often used in control systems or signal processing, commonly used to determine feedback whether the system is stable. Each point on the Nyquist plot corresponds to the frequency response at a specific frequency. The angle of the point relative to the origin represents the phase, and the distance from the origin represents the gain. Therefore, the Nyquist plot combines the amplitude and the Bode plots of the phases are combined in one graph.

Figure 4a–c shows the experimental results of the EIS in Bode and Bode-phase representations of the modified concretes considering period cure of 1, 7, 28 and 365 days.

Although Nyquist plots will forwardly be discussed, Figure 4d–f depict results in Nyquist representation of the examined sample (CC, SAND and CEM) after 1, 7, 28 and 365 days of immersion. The doted red lines mean the straight lines at 45° with $Z_{Real}$ axis (component), which induces porous electrode behavior as will be detailed. However, from this point, it can be said that the same trends are observed when Bode and Bode-phase are also indicated.

Remembering the CC sample is designated as that of the sample with conventional mixture (cement content of 368 kg/m$^3$) as shown in Table 1. The results of 1 day and 365 days were also evaluated (not included in mechanical analyses). The period of 1 day was included in EIS analyses due to it being expected that ions are forming double layers at the initial immersion period, constituting intermediates species. The oxide films were formed and possibly penetrated into the cement paste.

The period of 365 days was analyzed in order to verify the corrosion behavior trend after a long-term immersion period. Duplicates (i.e., sample #1 and #2) were provided for all examined samples considering all experimented periods of immersion. Examples of duplicates for the CC and CEM sample at 1 and 7 days are depicted in Figure 5. Duplicates were performed with twice experiments in a same sample examined; and also considering a same mixture in two distinctive specimens, i.e., in a different batch using same mixture.

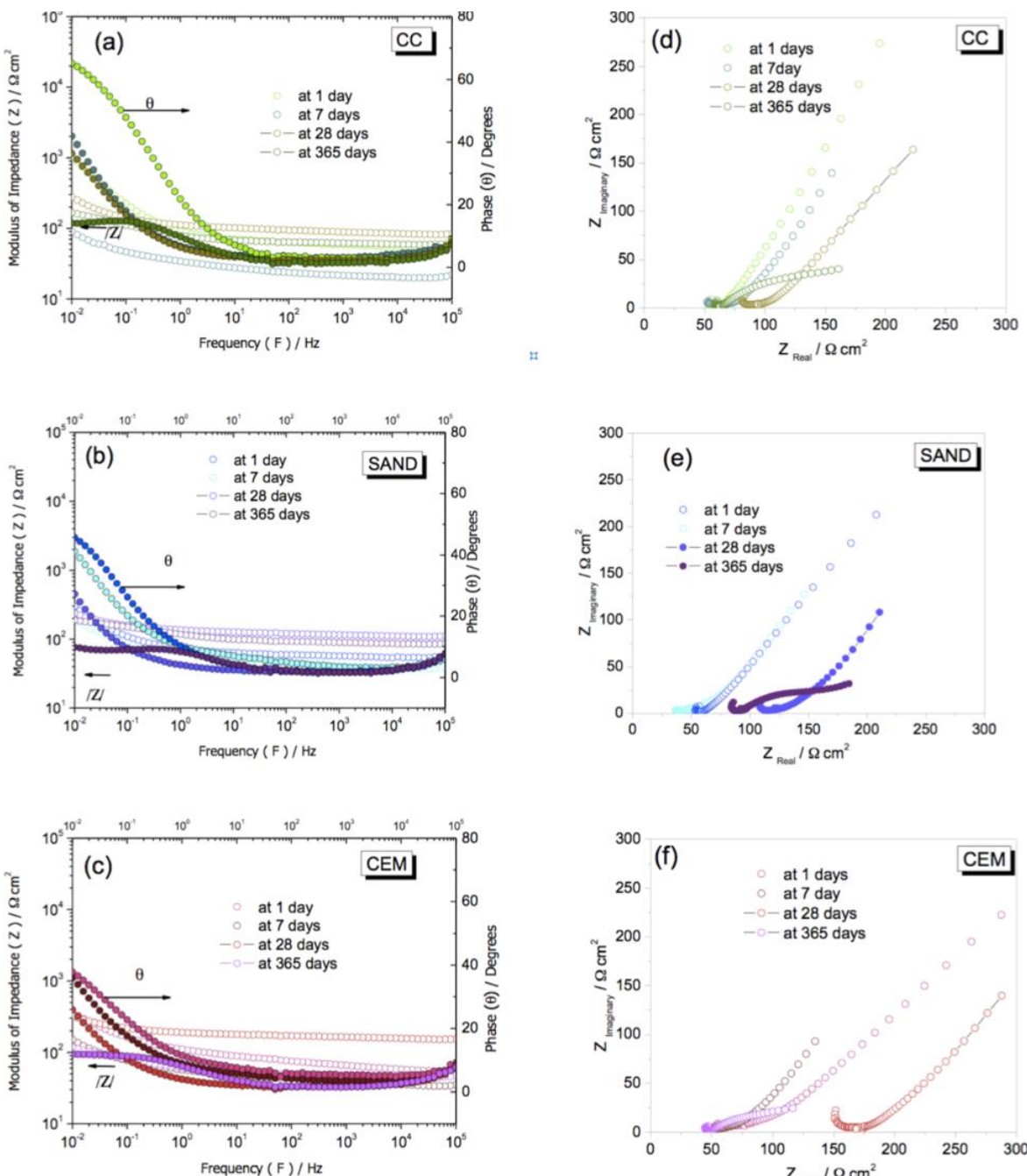

**Figure 4.** Experimental results of EIS of the CC, SAND and CEM samples at distinct curing/immersion curing periods (1, 7, 28 and 365 days) in a stagnant/naturally aerated 0.5 M NaCl solution. (**a–c**) depicts Bode and Bode-phase plots and (**d–f**) show Nyquist plots in different examined days.

Although few slight variations were observed, no modifications in the trends of the evaluated corrosion behavior are provided. Based on those experimented Bode-phase diagrams (Figure 4), it seems that only two time constants were prevalent. The first constant is characterized from high frequency domain up to an intermediate frequency domain (~1 Hz). Another time constant seems to be constituted at low frequency region (between $10^0$ Hz and $10^{-2}$ Hz) as depicted in Figures 4 and 5. However, these observations are simply speculative.

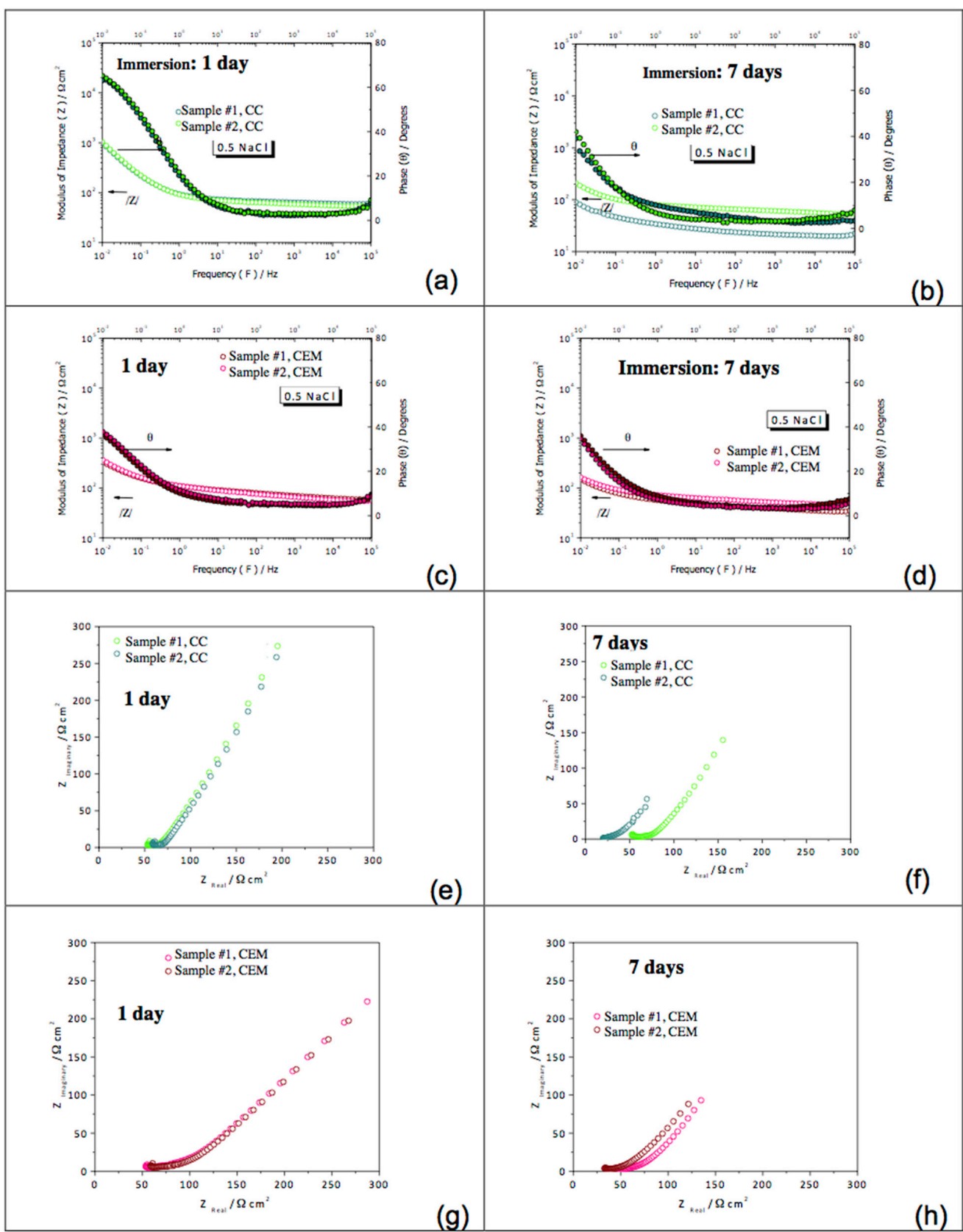

**Figure 5.** Duplicate of the results for the CC and CEM for 1 and 7 days in Bode/bode-phase (**a**) and (**b**), (**c**) and (**d**); respectively; and Nyquist plots in same periods for the CC and CEM samples for 1 day (**e**) and (**g**); and during 7 days during 1 and 7 days; (**f**) and (**h**), respectively.

A technique to consolidate this hypothesis is adopting De Levie's theory. Based on this, the number of time constants involved in a corrosive mechanism is characterized by a method proposed by Hirschorn and colleagues [67,68] and also other previously reported distinctive studies [59,69–72]. For this purpose, the moduli of the imaginary parts of the impedances vs. frequencies were analyzed as shown in Figure 6. Although only results corresponding with periods of 7 days are shown, all examined samples in all investigated immersion periods, only two time constants were characterized.

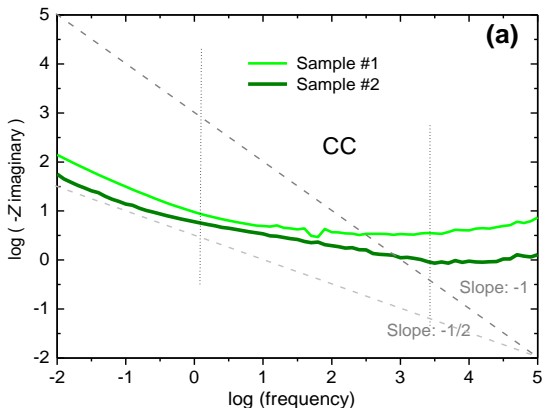
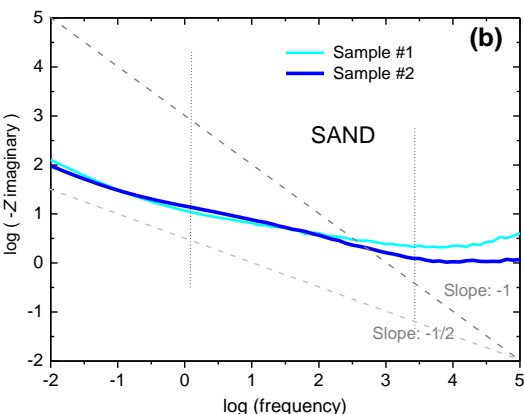

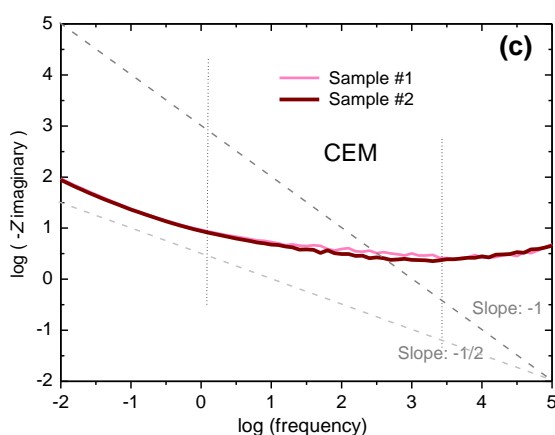

**Figure 6.** Moduli of the imaginary parts of the impedances vs. frequencies of: (**a**) the CC, (**b**) the SAND and (**c**) the CEM samples (Figure 6a–c, respectively) samples over 7 days of immersion in NaCl solution. Porous behavior is characterized when slopes are about −1/2 or lower (−1/4), while the planar electrode behavior has a slope of about −1.

Figure 6a shows the results of modulus of the imaginary part of the impedance vs. frequency of the CC sample (conventional/reference or control concrete sample). Duplicate results are shown for each examined sample. Figure 6b,c shows the results corresponding with the SAND and CEM samples, respectively. Considering the slopes constituted between $\log(-Z_{Imaginary})$ and $\log$ (frequency), between frequencies of about $10^3$ and $10^0$ Hz, all examined samples clearly exhibit slopes approximately −1/4 and at frequency domains lower than $10^0$ Hz. The slopes approximately of −1/2 were characterized. This demonstrates the participation of the electrode behavior in the corrosion process [45–48].

Considering distinctive materials, the aforementioned slopes corresponding with porous electrode behavior were previously reported in the literature [67–74]. When a planar electrode behavior is involved, the resulting slope is close to −1 [74], while a porous behavior can be associated with slopes of about −1/4 and −1/8, in some cases [74].

The porous electrode behavior electrode was demonstrated by Levie [45] and widely confirmed for distinctive systems [67–74]. Analyzing the Nyquist plot, this behavior occurred at a high frequency domain, and it was characterized by a Warburg-like behavior [45,67–73]. In a very simplified term [45,71–73], a porous electrode is shown in Equation (2) [45–48,67,68]:

$$Z = (R_0 Z_0)^{1/2} \, coth \, (L \sqrt{(R_0/Z_0)}) \tag{2}$$

where $R_0$ and $Z_0$ are the electrolyte resistance ($\Omega/cm$) and the interfacial impedance for one-unit length, and L means the length of each pore in cm [73]. Based on this aforementioned formulation, independently of the frequency, De Levie demonstrated that the angle with the real axis of Warburg impedance tends to 45° when L tends to a semi-infinite condition, and the *coth* term reaches 1. This indicates that Warburg impedance is half that reached by a flat electrode [45]. Macdonald [46] also discussed porous electrode behavior mainly describing equation and diffusion control in a semi-infinite pore, which clearly depicts a phase angle of $\pi/8$ with 22.5° defined.

Similarly, Bastidas [48] also reported a detailed study including interpretation of impedance data, which involves both porous electrode and diffusional phenomenon throughout a porous material. Interesting the distinct method of porous using De Levie, Park and Macdonald methods were compared and reported by Bastidas [48]. Murray [74] reported that when the ionic reaction is controlled, Bode magnitude versus frequency slopes of $-1/2$, $-1/4$ and $-1/8$ can also be observed. With these assertions, at high frequency domains in the Nyquist plots of the examined concrete samples, it is clearly characterized at 45° as depicted in Figure 4c–e.

These ranges forming 45° increase with the increase of the immersion periods for all examined concrete samples as also depicted in Figure 4d–f. The CC samples at 1 day of immersion characterized a minimal branch at 45° (in Nyquist plot, Figure 4d–f). This suggest that both the planar (conventional) and porous electrode behaviors were prevalent. When 45° is observed in Nyquist plots at low frequencies, the maximum angle θ reached about 40 degrees as shown in Figure 4.

During the last 20 years, similar behavior (porous behavior) has also been detected (in Nyquist plots, but not reported and discussed) in a great number of previous investigations considering corrosion of the steel bars embedded in concretes [33,38,39,46,58,75–78]. These referenced studies work distinct equivalent circuits and report adequately the resulting corrosion mechanisms containing various intermediates products (e.g., chlorides, oxygen and $CO_2$) and also demonstrating the coating interactions [78].

In a recent article, Kamde and Pillai [78] reported that at Nyquist plots; planar electrode behavior is characterized when coating (protecting steel bar) is not degraded. On the other hand, when the moisture penetrates throughout coating layer attacking steel bar, at a high frequency domain, the Nyquist plot clearly demonstrates an angle of 45° with $Z_{real}$ (x-axis). Additionally, when a steel bar promotes a rust layer formation, at a low frequency region, a Warburg component is characterized but forming a 22.5° with $Z_{Real}$ also confirming the porous behavior controls the corrosion mechanism as also reported by Macdonald and collaborators [46,47] and Bastidas [48]. Bastidas [48] also stated that a constant phase behavior of 22.5° with the real axis was constituted.

Figure 7a–c show the schematic representations of Nyquist plots with planar, porous and porous with Warburg component (transport and diffusion), respectively. These Figures are proposed based on previous reported investigations [37,38,59–61] and associated with corrosion mechanism phenomena proposed to occur. At high frequency domains, a small branch of Nyquist plot at 45° is characterized as demonstrated in Figure 7a. This indicates a predominant planar electrode behavior with a degradation of the embedded bar represented by a higher resistance $R_{Bar}$ as will be discussed. On the other hand, the predominant porous electrode behavior is depicted in Figure 7b,c.

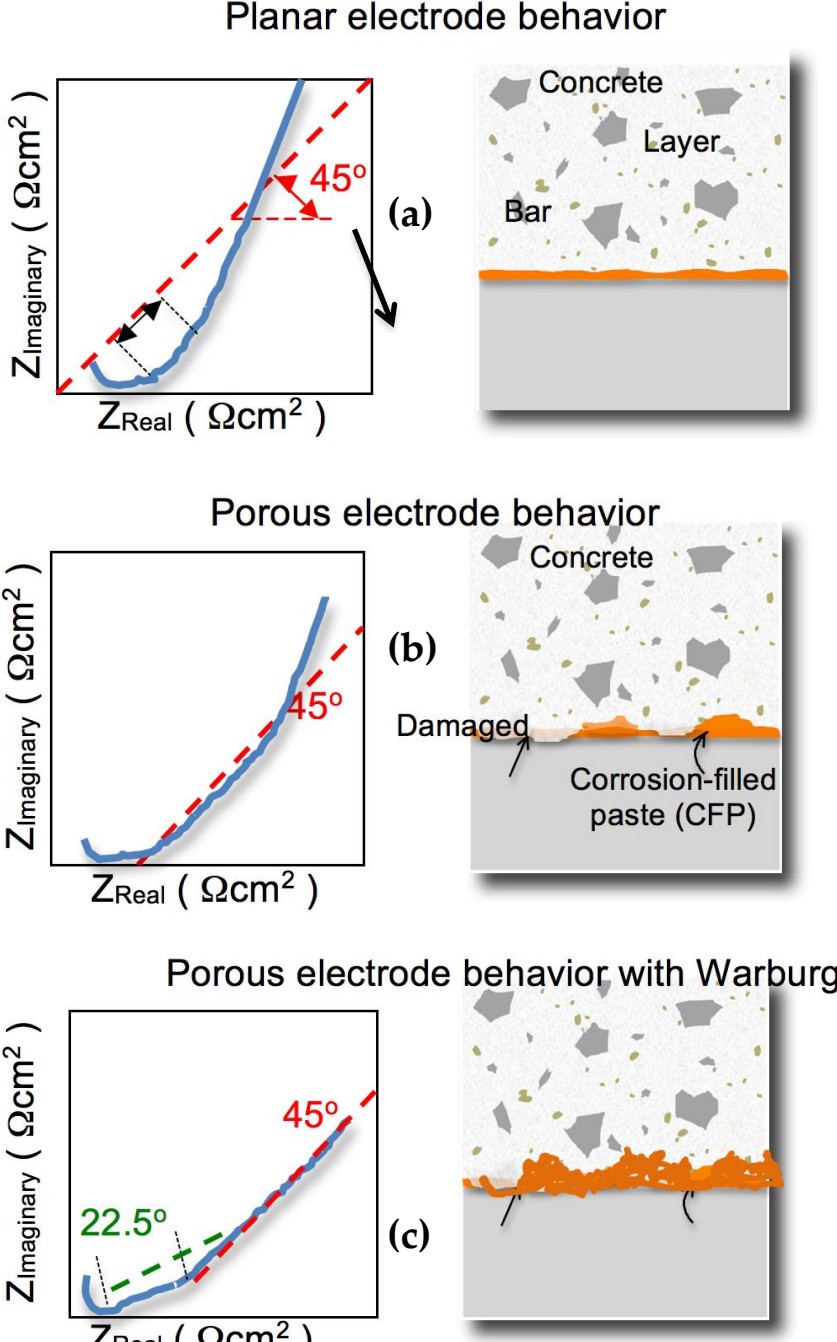

**Figure 7.** Schematic representations of the Nyquist plots and corresponding corrosion layer formation with predominant: (**a**) planar electrode behavior, (**b**) porous behavior and (**c**) porous with Warburg component formation.

A rust layer is also formed and, depending the constituents, a reasonable protective layer is constituted. Commonly, when the polarization curves are analyzed, a trend to passive behavior is observed. With increasing the immersion period, the ions migrate to concrete paste and regions designated as corrosion-filled paste (CFP) are formed as shown in Figure 7b. With the increase of the immersion period or aggressiveness level of corrosion, the rust layer is damaged and the CFP region is increased.

Thus, the steel bar seems to be locally degraded, and corrosion by-products are formed. With the corrosion evolution and depending on the imposed conditions, a CFP formation is constituted. Depending on the corrosion layer formed, a reasonable protection

is constituted. At high frequency domain, a 22.5° is formed, and a trending to straight line at 45° (Warburg) is also characterized as also previously reported [45–48,74] and depicted in Figure 7c.

Concerning to these aforementioned assertions, it is demonstrated that a porous electrode behavior domains the corrosion mechanism of the rebar embedded in the distinctive concrete. Based on Nyquist diagrams, it is qualitatively confirmed the corrosion behavior is intimately associated with interactions between rebar surface and electrolyte, electrolyte with cement paste and cement paste with rebar. This considering distinctive ions associated with chloride (electrolyte) penetrating throughout corrosion layer interacting with the rebar surface. This discussion will be provided. In the next section, the results of EIS measurements are analyzed.

### 3.3. EIS Analyses: Corrosion Performance among Distinctive Concrete Samples

When a qualitative analysis is provided, a comparison among EIS diagrams has not indicated a trend or an assertive conclusion based on both the moduli of impedance and sizes of semi arcs observed in Nyquist diagrams. These parameters seems to be relatively similar for all examined samples. When the immersion periods are considered, it seems that the increase of the immersion periods, the semi arcs of Nyquist are decreased and displaced to right side. This suggests that the increase of the immersion period, the rebar corrosion resistance is decreased as also previously reported [38]. We observed that semi arcs corresponding with 1 day of immersion were higher than 7 days.

The semi arcs related with 28 days were lower and were displaced when compared with 7 days. However, this has a rather and poor contribution to predict both corrosion mechanism and performance of rebar in distinct mixture of concrete. No conclusion can be attained concerning to replacements provided (i.e., sand and cement portions with EG particles). Based on this, two actions are required. A first is to determine the quantitative parameters of EIS results using an equivalent circuit.

Another is to analyze the potentiodynamic polarization curves of each one of the examined rebars embedded in distinctive concretes. This latter analysis contributes prior information obtained from EIS analysis by using their impedance parameters and distinctive mixture concretes. Initiating by the impedance parameters analyses, an equivalent circuit (EC) is adopted. This EC is written in a ZView software®, Scribner Ass. Inc., Southern Pines, NC, USA (version 2.1b) and a CNLS (complex non-linear least squares) simulation is performed. The selected EC is widely utilized in literature to prescribe steel bar corrosion in concrete [37–39,75–78]. In this present investigation, the element significances are assumed similarly to that recently reported (at 2020) by Sohail et al. [39] and also other investigations [76,78].

It is usual and accepted that, between concrete and rebar, the interfacial film and double layer are constituted [37–39,76,78]. Associated with these three physical characteristics, there are corresponding elements in EC. A resistance associated with a capacitance CPE (constant phase element) are related to the concrete bulk + electrolyte (NaCl penetrated), other resistance and capacitance associated with interface (between concrete and double layer). Additionally associated with steel bars are their corresponding resistance and capacitance (CPE). In addition, a Warburg element in series with resistance corresponding with charge transfer of the bar is attributed.

A schematic representation of the proposed EC considering concrete with distinctive mixture elements, interface (forming corrosion by-products) and steel bar is shown Figure 8. This is proposed based on previously reported circuits and adapting with proposed corrosion mechanism. $R_{Concr}$ is resistance of concrete associated with their capacitance $Q_{Concr}$. At interface between concrete and bar, there exists $R_{Interf}$ and their capacitance $Q_{Interf}$, corresponding with oxide layer. $R_{Bar}$ and $Q_{Bar}$ are the resistance and capacitance associated with charge transfer and double layer of the embedded steel bar. Finally, a Warburg element interfacing bar and oxide layer is proposed. This is associated with a possible diffusion and

transport occurring at this interface. No additional resistance $R_0$ (commonly at frequency range $10^6$ and $10^8$ Hz) is considered as also reported by Sohail et al. [39].

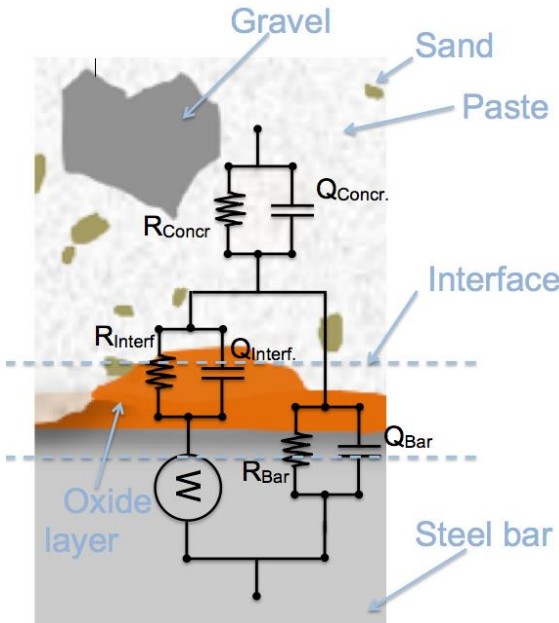

**Figure 8.** Proposed equivalent circuit (EC) utilized to simulate the EIS parameters.

Based on the consolidate fact that a porous electrode behavior is prevalent to predict the corrosion mechanisms of the embedded bars in the three distinct concretes, a schematic representation of initial and evolution of corrosion process associated with the three concrete mixtures is proposed. The representations for an initial immersion period of the three concretes are shown in Figure 9a–c. This is proposed based on phenomena typically occurring in corrosion systems and adapting with proposed mechanism for this investigation considering ES particles additions.

The penetrated water, oxygen and $Cl^-$ ions (from NaCl solution) interact with elements of concrete (constituents and phases, e.g., tricalcium aluminate $C_3A$ and calcium silicate hydrate C-S-H) and electrolyte (NaCl) and these with steel bar. Coarse and fine aggregates are also depicted. Although initial stage of immersion is designated, it is important to remember that concrete samples were immersed over 7 days in the curing period. Thus, the immersion period of 1 day corresponds with 1 day after their curing period. Similar occurs with immersion 28 days corresponding with 28 days after curing of 7 days.

Considering the initial immersion period, all samples formed double layers. Due to $CaCO_3$ additions and their interactions with cement paste, the cement hydration is slightly modified as previously reported [7,15–18,21]. This modification affects both pH and nature of double layer constituted. Matschei et al. [14] demonstrated that $CaCO_3$ particles affected the amount of free calcium hydroxide (C-H, $Ca(OH)_2$)) and C-S-H is unmodified.

They also stated that $CaCO_3$ acts with cement paste as filler or inert additional element. These interactions depend on the $CaCO_3$ size particles. Guo et al. [79] recently reported that $C_3A$ hydrated (e.g., $3CaO.Al_2O_3.6H_2O$) reacts with both chloride ions and $CaCO_3$ acidifying local limited cement paste and the chloride binding capacity decreased [79–81]. This can be simplified by Equations (3) and (4) (not stoichiometric balanced), respectively [79]:

$$4/3\,C_3A + 2Cl^- + 4H_2O \rightarrow 3CaO\cdot Al_2O_3\cdot CaCl_2\cdot 10H_2O + 2/3\,Al(OH)_3 + 2OH^- \qquad (3)$$

$$3CaO\cdot Al_2O_3\cdot CaCl_2\cdot xH_2O + Al(OH)_3 + CaCO_3 \rightarrow C_3A\cdot CaCO_3\cdot xH_2O + Cl^- \qquad (4)$$

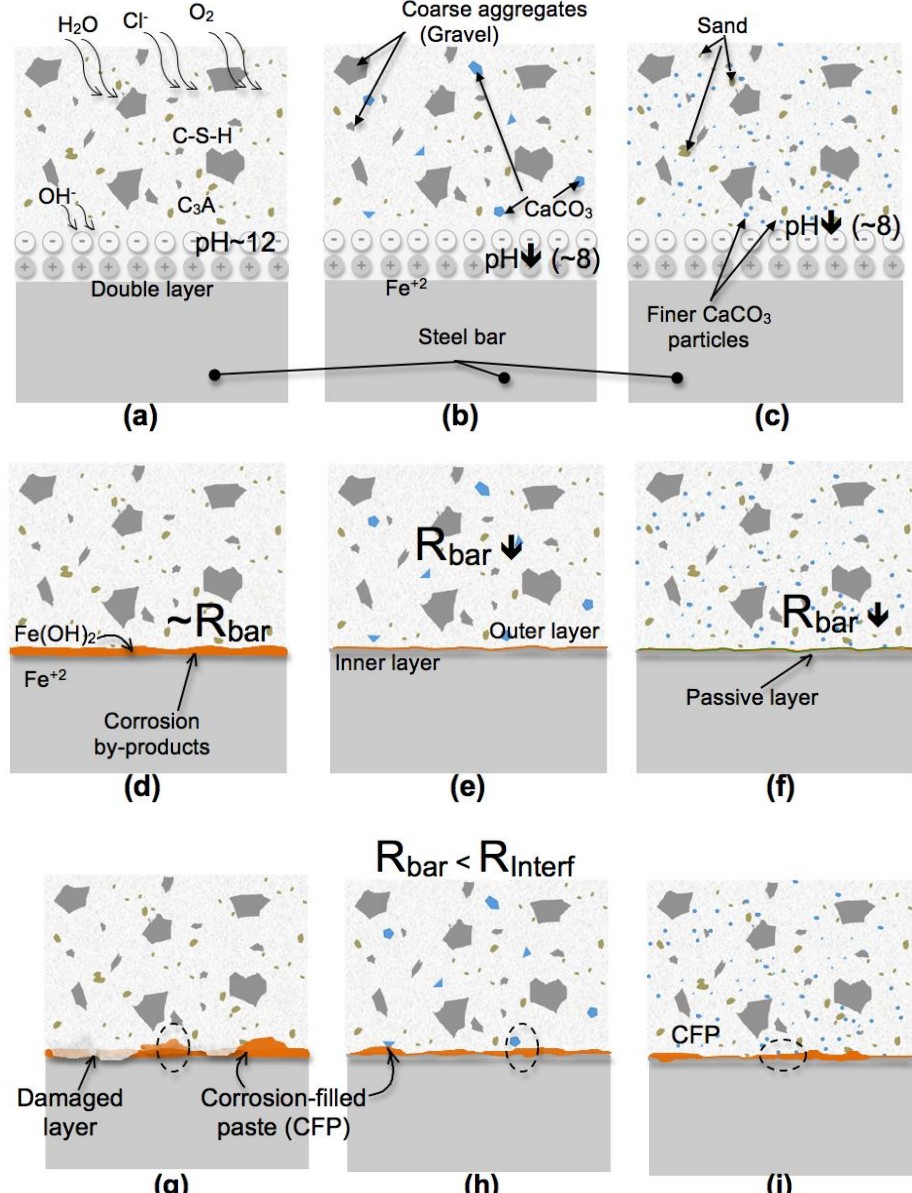

**Figure 9.** Schematic representation involving of the three distinct concretes: (**a**) the conventional concrete sample (CC), (**b**,**c**) the SAND and the CEM samples the SAND sample containing 10 wt.% EG particles, respectively, at initial stages of immersion. (**d**–**f**) sequential stages of immersion (e.g., 7 days) with evolution of double layer to corrosion (film) layer and interaction with bar, and (**g**–**i**) the relative long-term (~365 days) immersion with possible damaged oxide layer and corresponding corrosion-filled (CFP) paste.

In this initial stage, the $CaCO_3$ content modifies pH-neighboring rebar and different species to constitute the double layer are formed. With this, a passivation and resistance to chloride penetration is provided [79]. Considering a simplified Fe/water Pourbaix diagram, the corrosion and passivation domains corresponding to the CC sample and the SAND and the CEM samples were relatively different when compared. Considering that a pH of about 8 [40,75,80] is attained when $CaCO_3$ content is involved (instead pH ~12.5 [75,80]), at 1 day of immersion, the open circuit potentials (OCP) indicate that the CC samples has a potential of about −500mV (SCE). The SAND and CEM were of about −700mV (SCE). Their corresponding potentiodynamic polarization curves are shown and discussed. Based on a Pourbaix diagram, it is suggested that the CC sample locates in a corrosion domain

(pH ~12.5), while, at −700 mV(SCE) with pH = ~8, both the SAND and CEM samples were in the passivation domain as depicted in Figure 10.

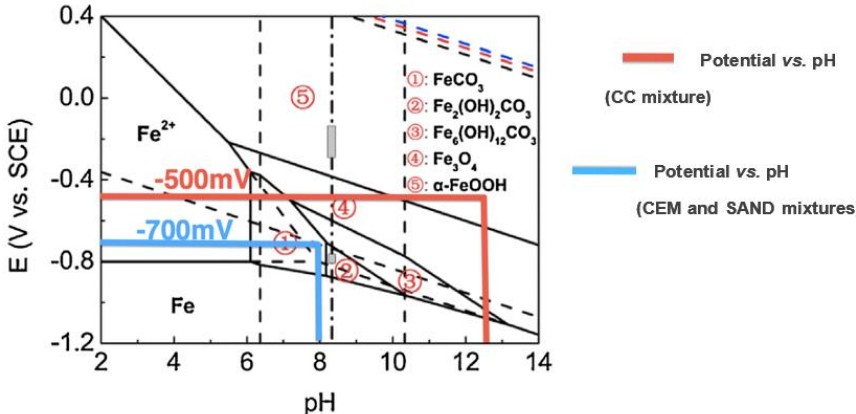

**Figure 10.** Pourbaix diagram of Fe-$H_2O$-$CO_2$ system adapted from Pourbaix, 1973, [81].

It is recognized that distinct species (solid, ionic and soluble) are involved when concrete paste is considered. With this, a potential-pH system of Fe-$H_2O$ [40,75] no correctly represents possible reactions. It is remarked that pH values neighboring rebar is extremely difficult since pH drop locally or punctually occurs. Due to this, no experimental measurements were conducted. A pH bulk measurement was not detected.

Considering all assertions previously provided, it is remembered that the proposed EC is used to impedance parameters be determined. With this, it is possible to compare the corrosion resistance behavior of the steel bars embedded. Since it is understood that $CaCO_3$ content provokes a local drops in pH and possibly a passivation is attained, the impedance parameters using an EC (shown in Figure 8) over 1 and 7 days and 28 and 365 days of immersion periods are organized in Tables 3 and 4, respectively. Potentiodynamic polarization curves associated with these distinct immersion periods are also discussed. In a general way, considering Tables 3 and 4, we observed that all examined samples have capacitances $Q_{Bar}$ higher than $Q_{Interf}$ for all immersion periods examined.

**Table 3.** Impedance parameters determined using equivalent circuit (EC) shown in Figure 8 with steel bars embedded in distinct concretes over 1 and 7 days.

| Parameters | CC 1 Day | CEM 1 Day | SAND 1 Day |
|---|---|---|---|
| $Q_{Concr}$ ($10^{-12}$ F/cm²) | 3228 (±64) | 11.2 (±2.7) | 5.4 (±0.2) |
| $R_{Concr}$ (Ω cm²) | 54 (±0.2) | 42.3 (± 1) | 46.9 (±0.1) |
| | | | |
| $Q_{Interf}$ ($10^{-3}$ F/cm²) | 0.6 (±0.02) | 1.5 (±0.02) | 1.1 (±0.03) |
| $R_{Interf}$ (Ω cm²) | 14.2 (±0.5) | **45.4 (±7)** | 14 (±0.1) |
| $n_2$ | 0.63 | 0.33 | 0.38 |
| | | | |
| $Q_{Bar}$ ($10^{-3}$ F/cm²) | 4.9 (±0.5) | 3.9 (±0.2) | 12.1 (±0.1) |
| $R_{bar}$ (Ω cm²) | **130 (±18)** | 47.3 (±1.2) | 32.6 (±3) |
| $n_3$ | **0.75** | 0.54 | 0.61 |
| | | | |
| $W$ (Ω cm²) | 7543 (±145) | 2938 (±20) | 1332 (±11) |
| | | | |
| $\chi^2$ | $8.2 \times 10^{-4}$ | $1.5 \times 10^{-4}$ | $5.5 \times 10^{-4}$ |
| Σ Sqr. | 0.09 | 0.02 | 0.08 |

**Table 3.** *Cont.*

| Parameters | CC 7 Days | CEM 7 Days | SAND 7 Days |
|---|---|---|---|
| $Q_{Concr}$ ($10^{-12}$ F/cm$^2$) | 3.8 ($\pm$0.8) | 0.5 ($\pm$0.08) | 0.3 ($\pm$0.08) |
| $R_{Concr}$ ($\Omega$ cm$^2$) | 42.9 ($\pm$0.1) | 15.7 ($\pm$0.02) | 16.9 ($\pm$0.03) |
| | | | |
| $Q_{Interf}$ ($10^{-3}$ F/cm$^2$) | 3.0 ($\pm$0.03) | 4.1 ($\pm$0.02) | 3.8 ($\pm$0.03) |
| $R_{Interf}$ ($\Omega$ cm$^2$) | **20.8($\pm$0.1)** | **25.6 ($\pm$0.08)** | **25.8 ($\pm$0.13)** |
| $n_2$ | 0.35 | 0.32 | 0.30 |
| | | | |
| $Q_{Bar}$ ($10^{-3}$ F/cm$^2$) | 16.5 ($\pm$0.1) | 10.6 ($\pm$0.07) | 6.9 ($\pm$0.6) |
| $R_{bar}$ ($\Omega$ cm$^2$) | 132 ($\pm$1.5) | 24.4($\pm$0.2) | 34.7 ($\pm$0.4) |
| $n_3$ | 0.54 | 0.48 | 0.48 |
| | | | |
| $W$ ($\Omega$ cm$^2$) | 44.5 ($\pm$5) | 382 ($\pm$7) | 546 ($\pm$11) |
| | | | |
| $\chi^2$ | $2.3 \times 10^{-4}$ | $1.4 \times 10^{-4}$ | $3.3 \times 10^{-4}$ |
| $\Sigma$ Sqr. | 0.03 | 0.02 | 0.05 |

Additionally, over 1 day and 7 days, the values of $R_{Bar}$ were also higher than $R_{Interf}$ for all samples examined. This indicates that $R_{Bar}$ predominantly dominates the corrosion mechanism. This is mainly when the resistances and capacitances were concomitantly evaluated. Concerning to those values reported in Table 3, when values corresponding with 1 day of immersion for all examined samples were observed, the $R_{Concr}$ values have the same order of magnitudes as expected. The highest $R_{Bar}$ is that of the reference (CC) sample (~130 $\Omega$cm$^2$). The same orders of magnitudes were observed when both the SAND and CEM samples were analyzed.

When the $R_{Interf}$ values were compared, the CC and SAND samples are ~14 $\Omega$cm$^2$, while the CEM sample is ~45 $\Omega$cm$^2$. The CEM sample has similar values for $R_{Interf}$ and $R_{Bar}$ (~45 $\Omega$cm$^2$). This induces that at initial immersion period, at interface between bar and concrete + electrolyte of the CEM sample, a different reaction occurs when compared with other two concrete samples. Associating these impedance parameters with those potentiodynamic polarization curves, we observed that a passivation occurs for the CEM sample as depicted in Figure 11.

When the results for 7 days of immersion period were analyzed, we observed that $R_{Bar}$ of the CC sample has not substantially modified (~130 $\Omega$cm$^2$). However, the corresponding $R_{Bar}$ values of the SAND and CEM samples decreased, while $R_{Interf}$ were similar for all samples examined (between 20 and 25 $\Omega$cm$^2$). Ghorbani et al. [38] also attained similar observation when steel bars embedded in modified concretes (with marble and granite) over 14, 28 and 90 days were evaluated. Comparing the results for 1 day and 7 days, at 1 day the CEM sample has higher $R_{Interf}$ than the CC and SAND samples. This seems to be associated with finer ES particles replacing with 10% wt.% cement content.

The potentiodynamic polarization curve of the CEM sample also reveals a distinctive behavior when compared to other two samples examined. A primary passive current density ($i_{pp}$) at 1 day is clearly characterized for both the sample #1 and #2 (duplicate) for the CEM sample. This $i_{pp}$ (of about $10^{-4}$ A cm$^{-2}$) occurs between at potential of about $-700$ and $-550$ mV (SCE). This behavior did not occur in other examined samples, which corroborates with those values of both the $R_{Interf}$ and $R_{Bar}$ corresponding with the CEM sample.

The passivation behavior occurs for both the CC and SAND samples only at 7 days of immersion. The CEM sample retains its passivation behavior, with a lower $i_{pp}$ than other samples as also observed in Figure 11. The experimental results for all examined

samples at 28 days are shown in Table 4. We verified that $R_{Interf}$ values were higher than $R_{Bar}$. Concerning to capacitances $Q_{Interf}$, these were still lower than $Q_{Bar}$.

**Table 4.** Impedance parameters determined using equivalent circuit (EC) shown in Figure 8 with steel bars embedded in distinct concretes over 28 and 365 days.

| Parameters | CC 28 Days | CEM 28 Days | SAND 28 Days |
|---|---|---|---|
| $Q_{Concr}$ ($10^{-12}$ F/cm$^2$) | 10.5 ($\pm$2.3) | 48.5 ($\pm$9) | 12.1 ($\pm$3) |
| $R_{Concrete}$ ($\Omega$ cm$^2$) | 39.2 ($\pm$0.1) | 79.6 ($\pm$0.08) | 63.5 ($\pm$0.08) |
| | | | |
| $Q_{Interf}$ ($10^{-6}$ F/cm$^2$) | 6.2 ($\pm$0.02) | 2.9 ($\pm$0.04) | 3.2 ($\pm$0.03) |
| $R_{Interf}$ ($\Omega$ cm$^2$) | 60.8 ($\pm$0.2) | **99.4 ($\pm$0.3)** | 59.3 ($\pm$0.3) |
| $n_2$ | 0.21 | 0.22 | 0.24 |
| | | | |
| $Q_{Bar}$ ($10^{-3}$ F/cm$^2$) | 6.4 ($\pm$0.1) | 9.4 ($\pm$0.1) | 9.3 ($\pm$0.1) |
| $R_{Bar}$ ($\Omega$ cm$^2$) | 16.3 ($\pm$0.6) | **27.2 ($\pm$1.3)** | 23.2 ($\pm$0.7) |
| $n_3$ | 0.60 | 0.58 | 0.52 |
| | | | |
| $W$ ($\Omega$ cm$^2$) | 721.7 ($\pm$14) | **831.5 ($\pm$4)** | 451 ($\pm$3) |
| | | | |
| $\chi^2$ | $2.5 \times 10^{-4}$ | $2.3 \times 10^{-4}$ | $3.7 \times 10^{-4}$ |
| $\Sigma$ Sqr. | 0.04 | 0.03 | 0.05 |

| Parameters | CC 365 Days | CEM 365 Days | SAND 365 Days |
|---|---|---|---|
| $Q_{Concr}$ ($10^{-12}$ F/cm$^2$) | 0.47 ($\pm$0.4) | 0.71 ($\pm$0.2) | 0.79 ($\pm$0.3) |
| $R_{Concr}$ ($\Omega$ cm$^2$) | 27.9 ($\pm$0.04) | 21.3 ($\pm$0.05) | 40.2 ($\pm$0.1) |
| | | | |
| $Q_{Interf}$ ($10^{-3}$ F/cm$^2$) | 0.021 ($\pm$0.003) | 0.053 ($\pm$0.001) | 0.006 ($\pm$0.001) |
| $R_{Interf}$ ($\Omega$ cm$^2$) | 34.5 ($\pm$0.2) | 27.9 ($\pm$0.08) | 48.3 ($\pm$0.2) |
| $n_2$ | 0.45 | 0.42 | 0.52 |
| | | | |
| $Q_{Bar}$ ($10^{-3}$ F/cm$^2$) | 18.1 ($\pm$0.8) | 24.3 ($\pm$0.3) | 12.2 ($\pm$0.1) |
| $R_{Bar}$ ($\Omega$ cm$^2$) | 67.9 ($\pm$1.5) | **78.5 ($\pm$2)** | 6.4 ($\pm$3) |
| $n_3$ | 0.49 | 0.51 | 0.41 |
| | | | |
| $W$ ($\Omega$ cm$^2$) | 113 ($\pm$3) | 21.3 ($\pm$2) | 147 ($\pm$15) |
| | | | |
| $\chi^2$ | $2.4 \times 10^{-4}$ | $6.6 \times 10^{-4}$ | $6.5 \times 10^{-4}$ |
| $\Sigma$ Sqr. | 0.03 | 0.09 | 0.09 |

The fact that $R_{Bar}$ is lower than $R_{Interf}$ seems to be associated with initial condition verified for each one of the examined samples. At 28 days, the CC sample has its $R_{Bar}$ decreased of about 10 times when compared with 7 days. The values of $R_{Bar}$ corresponding with the CEM and SAND samples were not substantially modified when compared at 7 days. Interestingly, the CEM sample shows higher $R_{Bar}$ (of about 30 $\Omega$cm$^2$ against ~16 and 20 $\Omega$cm$^2$) and $R_{Interf}$ (of about 100 $\Omega$cm$^2$ against ~60 $\Omega$cm$^2$) than the CC and SAND samples. This indicates that at this period (28 days), the CEM sample has better corrosion resistance.

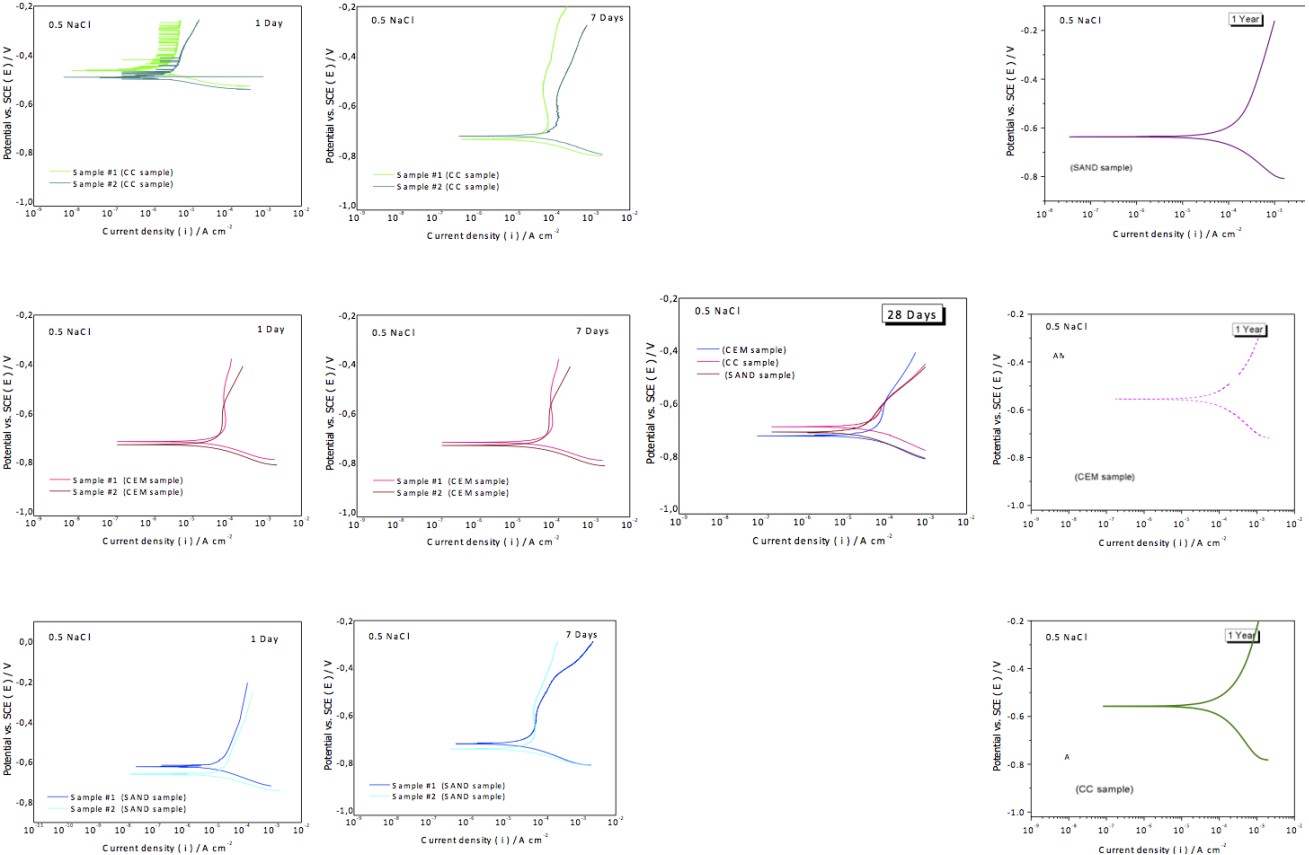

**Figure 11.** Experimental results of potentiodynamic polarization curves in NaCl for all examined samples in NaCl solution considering distinctive immersion periods (1 days, 7, 28 and 365 days).

When the EIS results of the period of 365 days were analyzed, similar observations at 1 day and 7 days were also verified for 365 days, i.e., $Q_{Bar}$ lower than $Q_{Interf}$ and majority $R_{Bar}$ higher $R_{Interf}$. Evidently, not only one immersion period should be considered to predict the resulting corrosion behavior. Additionally, not only the impedance parameters obtained from EIS measurements should also be taken in account. Some parameters obtained from the potentiodynamic polarization curves also have important roles in predicting the corrosion behavior and their evolution throughout distinctive immersion periods. Based on this, the corrosion current density ($i_{corr}$), their corresponding corrosion potential ($E_{corr}$) and $i_{pp}$ values were determined as shown in Table 5.

**Table 5.** Polarization and impedance parameters of the CC, SAND and CEM samples in distinctive immersion periods.

| Mixture | Parameter | Period (in Days) | | | |
|---|---|---|---|---|---|
| | | **1** | **7** | **28** | **365** |
| **CC** | $i_{corr}/\mu Acm^{-2}$ | **2.9 ($\pm$0.3)** | 22.3 ($\pm$0.4) | 19.4 ($\pm$0.5) | 88.0 ($\pm$2) |
| | $E_{corr}/mV$ | $-493$ ($\pm$2) | $-735$ ($\pm$2) | $-664$ ($\pm$2) | $-557$ ($\pm$1) |
| | $i_{pp}/\mu Acm^{-2}$ | - - - | 81 | 45 | - - - |
| | $R_{bar}/\Omega\ cm^2$ | **130 ($\pm$18)** | **132 ($\pm$1.5)** | 16.3 ($\pm$0.6) | 67.9 ($\pm$1.5) |
| | $R_{int}/\Omega\ cm^2$ | 14 ($\pm$0.5) | 21 ($\pm$0.1) | 61 ($\pm$0.2) | 35 ($\pm$0.2) |
| | $W/\Omega\ cm^2$ | 7.5k ($\pm$0.14k) | 44.5 ($\pm$5) | 721.7 ($\pm$14) | 113 ($\pm$3) |

**Table 5.** *Cont.*

| Mixture | Parameter | Period (in Days) | | | |
|---|---|---|---|---|---|
| | | 1 | 7 | 28 | 365 |
| SAND | $i_{corr}/\mu Acm^{-2}$ | 10.4 (±0.4) | 18.8 (±0.4) | 23.5 (±0.4) | 76.0 (±2) |
| | $E_{corr}/mV$ | −622 (±2) | −743 (±2) | −679 (±2) | −637 (±1) |
| | $i_{pp}/\mu Acm^{-2}$ | - - - | 65 | 71 | - - - |
| | $R_{bar}/\Omega\ cm^2$ | 32.6 (±3) | 34.7 (±0.4) | 23.2(±0.7) | 6.4 (±3) |
| | $R_{int}/\Omega\ cm^2$ | 14 (±0.1) | 26 (±0.1) | 60 (±0.5) | 48 (±0.2) |
| | $W/\Omega\ cm^2$ | 1.3k (±0.01k) | 546 (±11) | 451(±3) | 147 (±15) |
| CEM | $i_{corr}/\mu Acm^{-2}$ | 9.2 (±0.5) | 23.9 (±0.5) | 21.2 (±0.5) | **71** (±2) |
| | $E_{corr}/mV$ | −673 (±2) | −729 (±2) | −673 (±2) | −555 (±1) |
| | $i_{pp}/\mu Acm^{-2}$ | **39** | 96.6 | 54 | - - - |
| | $R_{bar}/\Omega\ cm^2$ | 47.3 (±1.2) | 24.4 (±0.2) | **27.2(±1.3)** | **78.5 (±2)** |
| | $R_{int}/\Omega\ cm^2$ | 45 (±7) | 26 (±0.1) | 99 (±0.3) | 28 (±0.1) |
| | $W/\Omega\ cm^2$ | 2.9k (±0.02k) | 382 (±7) | 831.5(±4) | 21.3 (±2) |

Although distinctive magnitude values of both $i_{corr}$ and $E_{corr}$ were reached during distinct immersion periods, when these values were analyzed, trends were attained as shown in Figure 12. At initial immersion (1 day), the $i_{corr}$ rapidly increases with a trend to stabilize (up to 28 days) and finally, at more long-term immersion period, the measured $i_{corr}$ values were considerably increased as depicted in Figure 12a.

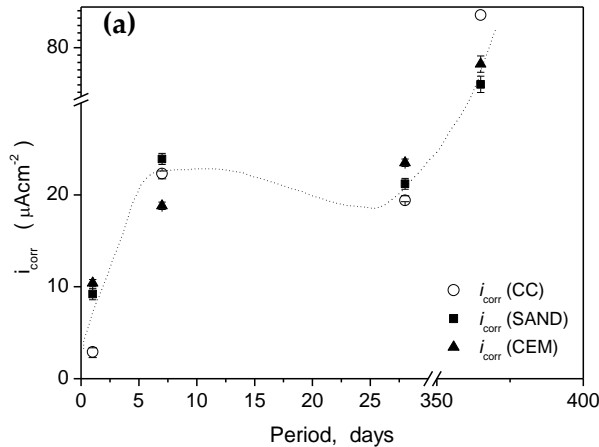

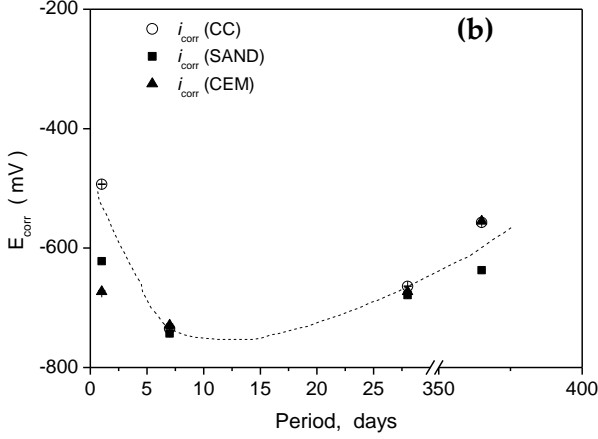

**Figure 12.** Variation of experimental values of: (**a**) $i_{corr}$ and (**b**) $E_{corr}$ with immersion periods of the three examined samples in NaCl solution.

Associated with the observed $i_{corr}$ trend during the immersion period, the $E_{corr}$ with immersion period similarly varies for all samples examined. At initial immersion, the $E_{corr}$ decreased (i.e., displaced to a more active potential region) with the increase of the immersion period, while $i_{corr}$ considerably increased as shown in Figure 12b.

*3.4. SEM Micrograph Observations*

Based on the aforementioned assertions, EIS and polarization are useful to predict the corrosion behavior considering both initial (short-term) and long-term immersion periods. Although these measurements induce to a selective concrete, there are other parameters to be considered. These will be discussed. Before, the resulting micrographs of the three samples were analyzed. Figure 13a,b show the longitudinal images of the CC, CEM and SAND samples and detailed image of the SAND sample demonstrating the ES particles homogeneously distributed throughout the concrete paste.

These samples were obtained after 365 days of immersion in NaCl solution. Figure 13c reveals a typical location at interface between steel bar and cement paste. Figure 13d,e show typical SEM images of the sample at interface steel/cement in distinct magnifications. The ES and sand particles were clearly characterized. Figure 13f shows EDX characterization and it helps to identify the sand particles and C-S-H and $C_3A$ phases.

The distinctive energy peaks of EDX patterns indicate (duplicate: Samples #1 and #2) that both Fe and Cl ions migrated in cement paste to constitute the corrosion-filled (CFP) paste region. Figure 14a,b are presented to demonstrate the resulting micrograph of the CEM sample. Both SE-secondary electrons and BSE-back scattering electron techniques are depicted. This sample has a portion of cement (10 wt.%) replaced with the fine ES particle content. These fine ES particles (10 wt.%) have a fineness modulus close to cement.

It appears that fine ES content reacts as filler as previously reported [12,14,21,27–29,32]. The phases C-S-H and $C_3A$ were also characterized as depicted in Figure 14c. It seems that the $C_3A$ particles were relatively lower (~60 μm) than those characterized when the SAND sample is analyzed (~300 μm). This observation seems to be also correlated with mechanical behavior reached. Although the cement content is decreased, the resulting mechanical behavior did not substantially decrease as aforementioned. As also expected, both Fe and Cl ions were also identified. However, its corresponding intensity peaks were lower than the SAND sample. This induces that ions penetrated the cement constituting CFP but showing a lower quantity when compared with the SAND sample.

Figure 15a–d show SEM images in two magnifications, the elemental map image and the resulting EDX patterns of the CC (reference or control) sample, respectively. We observed that main elements of the examined concrete were characterized, i.e., sand particle, and C-S-H and $C_3A$ phases. The $C_3A$ phase has resulting size similar to the CEM sample. Both Fe and chloride ions were also identified. These were similar to the CEM sample, which seems to be lower than the examined SAND sample.

Ming and Shi [82] recently demonstrated the corrosion layer and corrosion-filled paste after long-term immersion at steel/concrete interface. We found that an excessive corrosion products formation provokes crack propagation from steel bar to paste. These corrosion products penetrate through the cracks formed and CFP regions were increased and/or enlarged. They [82] also demonstrated that distinctive steel bar protections were provided with different corrosion products, when ions were involved due to different steel (Cr-rich) were investigated [39,42,82].

The corrosion layer thick depends mainly of the ions involved, immersion period and aggressive of electrolyte. Commonly, the corrosion layer thick varies between some few micrometers to some hundred micrometers as previously investigated [38,39,42,43,82]. As also previously reported [38–40,42–44,82], Ca and Fe contents at steel/concrete interface to CFP region were varied. Fe content increases at CFP region indicating their migration and resulting crack propagation.



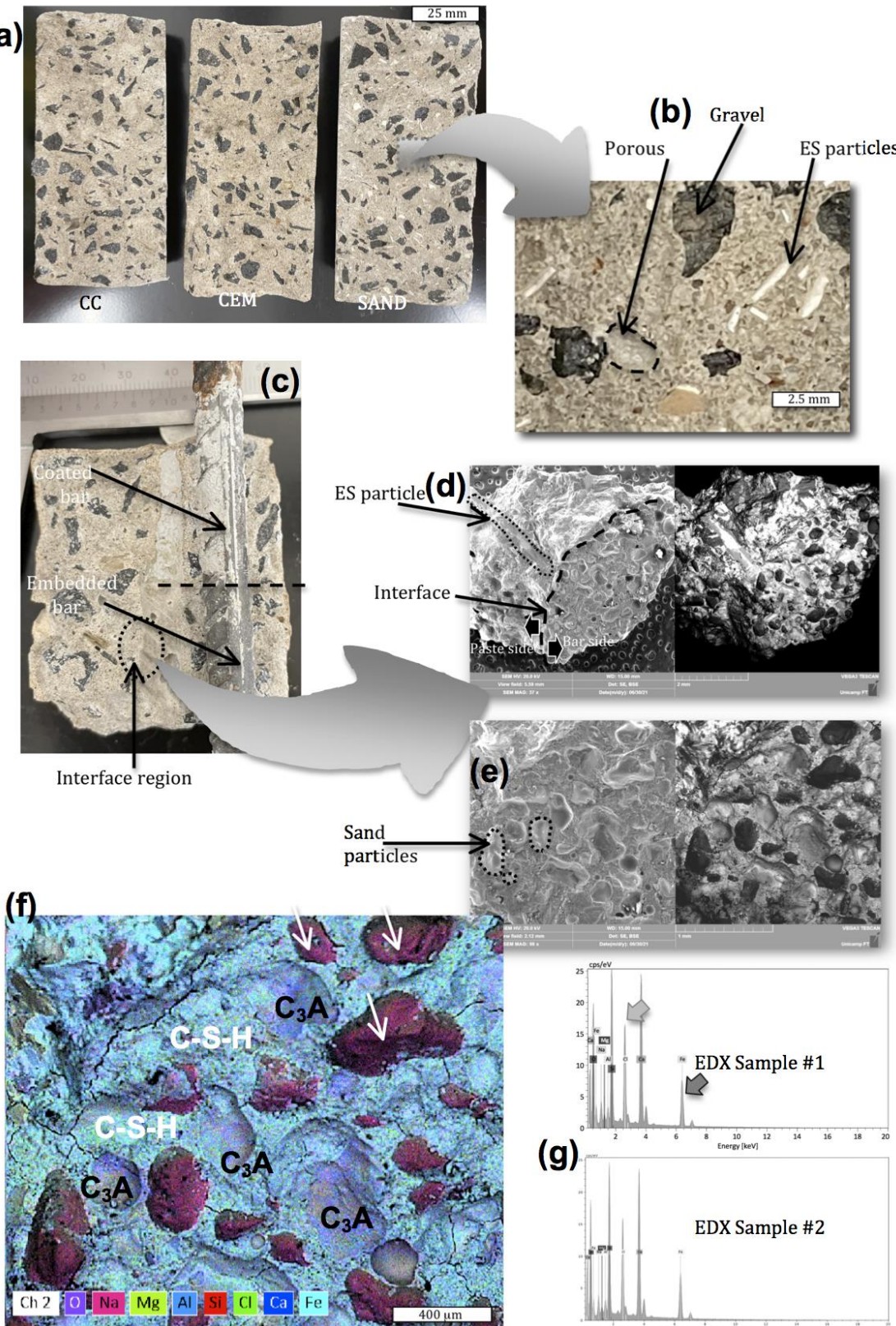

**Figure 13.** (**a**) The CC, CEM and SAND sample depicting the concrete microstructure, (**b**) showing elements constituting concrete including ES (egg-shell) particles, (**c**) demonstrating the local at interface bar/concrete where the samples were withdrawn, (**d,e**) SEM images obtained at interface bar/concrete, and (**f,g**) elemental map and EDX diagrams of the SAND sample.

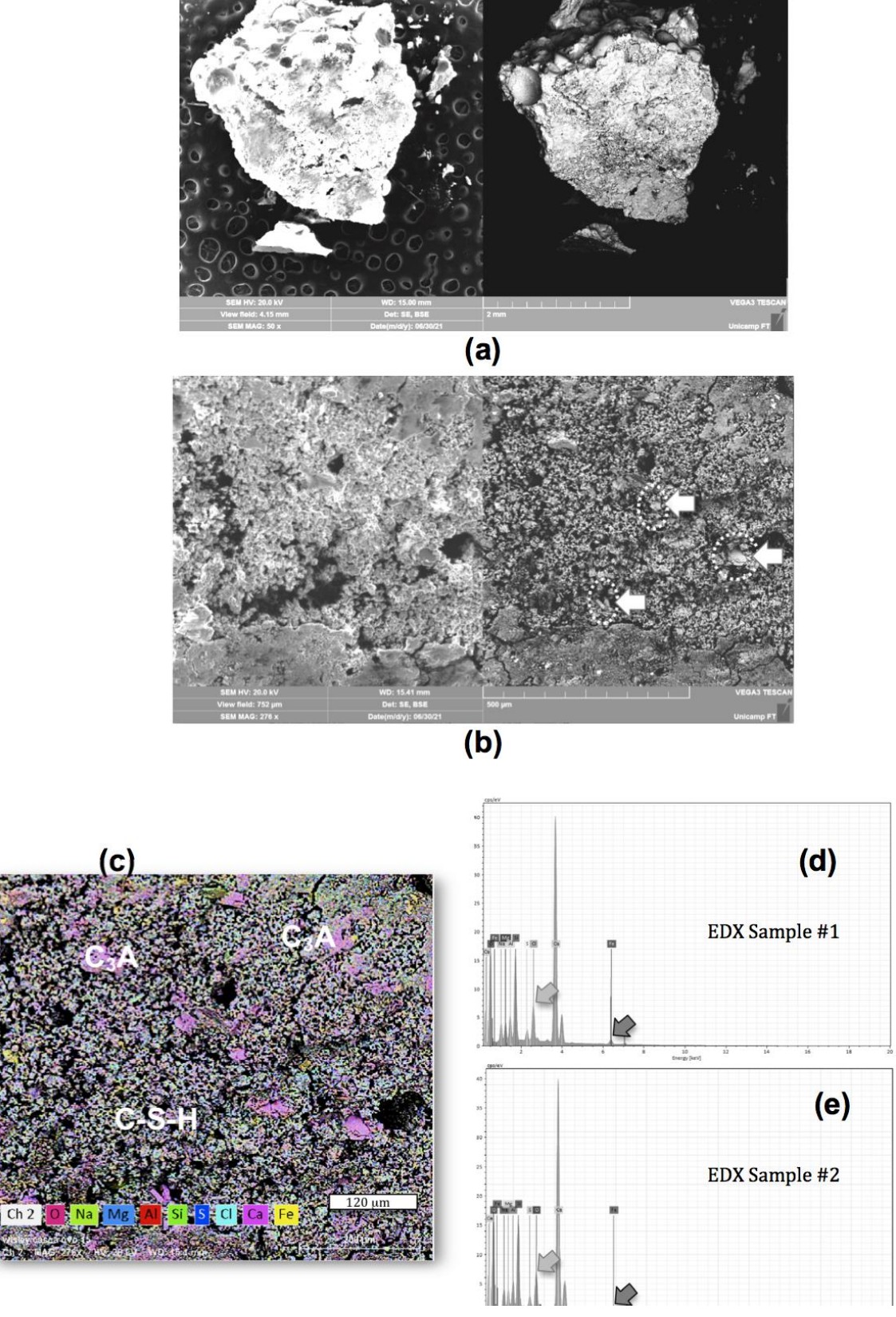

**Figure 14.** (**a**,**b**) typical SEM micrographs of the CEM sample and (**c–e**) elemental map and EDX diagrams (duplicate) of the CEM sample, respectively. Eggshell particles are identified by white arrows inside (**b**).

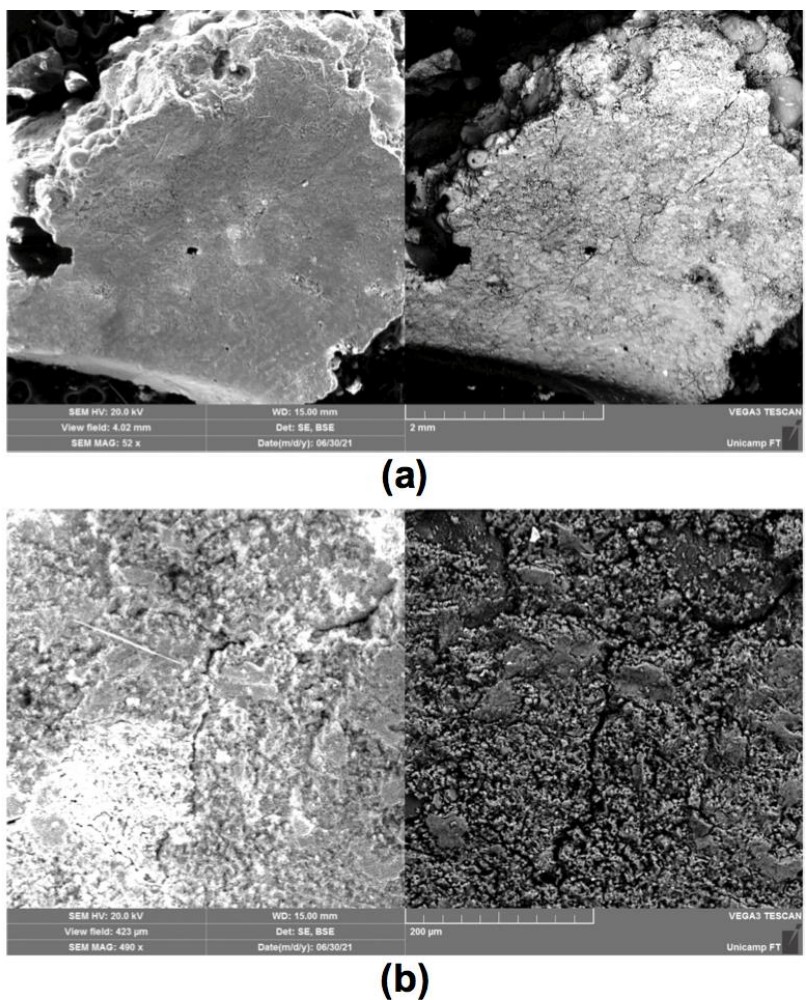

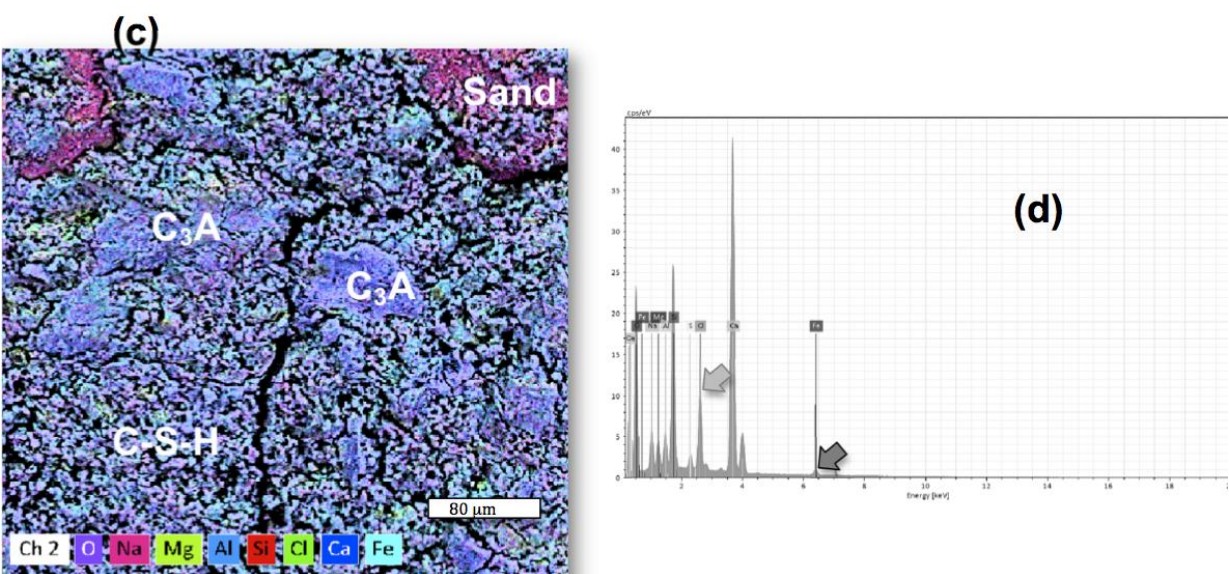

**Figure 15.** (**a**,**b**) typical SEM micrographs of the CEM sample and (**c**,**d**) elemental map and EDX diagram of the CC (reference/control) sample, respectively.

Distinct species FeOOH (goethite, akageneite and lepidocrocite, α, β and γ respectively.) and $Fe_2O_3$ and/or $Fe_3O_4$ were detected [40,42]. Other species can also be involved, such as $Fe_xO_y$ and $Fe_xCl_y$ as reported by Kamde and Pillai [78]. A schematic representation at interface steel bar/concrete, showing corrosion layer and Fe and Ca concentrations variations is shown in Figure 16a. A representation with coarse ES particles distributed throughout the cement paste is shown in Figure 16b.

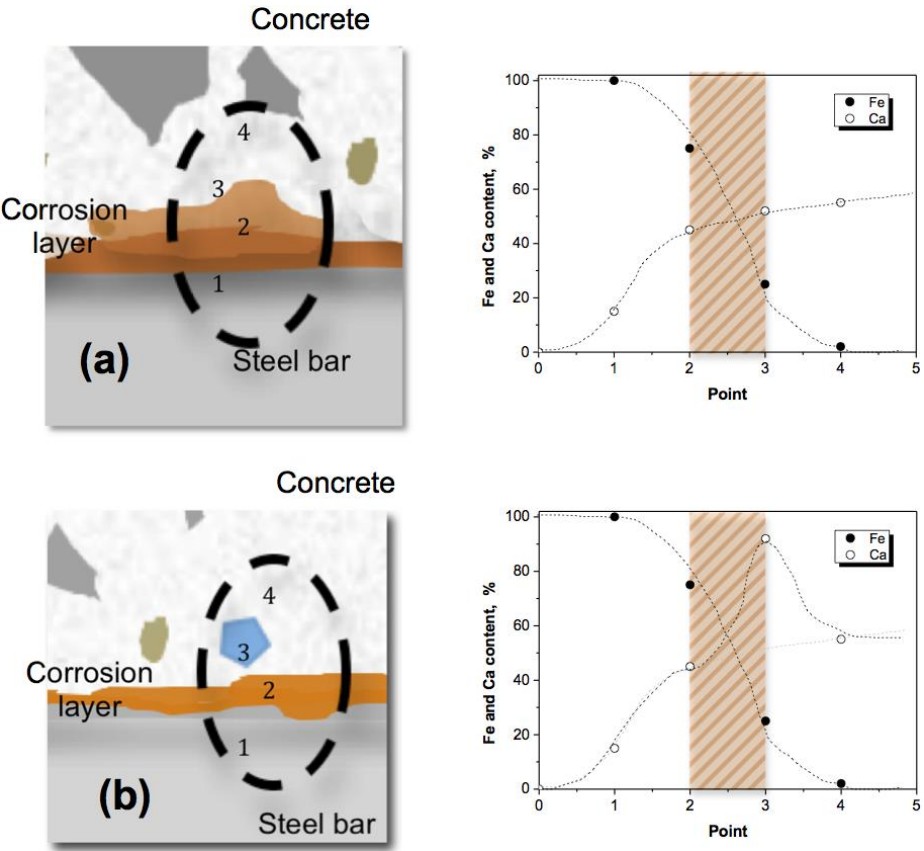

**Figure 16.** Schematic representation of Fe and Ca ions at transversal section between bar and concrete containing: (**a**) corrosion layer and corrosion-filled paste and (**b**) containing ES particle. Points #1, 2, 3 and 4 inside (**a**,**b**) corresponding with points of the right part of the figure.

We concluded that the corrosion responses of the three embedded steel bars in distinct concretes were modified with immersion period and concrete composition. In order to predict the corrosion behavior of rebar for each one of the examined periods, EIS and polarizations results must be analyzed. The water-to-cement for all examined concretes were parameterized. This parameter associated with the consumption of cement (Co) constitutes a limiting factor of the present investigation. Modification in the "Co", the compressive strength (CS) can slightly be modified. Based on this, the Co per CS ratios for the three examined concretes were determined as demonstrated in Table 6.

As previously reported [50], the values of Co/CS per CS indicate adequate results, i.e., the Co/CS decreases with the curing period and CS is increased. The lowest Co/CS is that of the CC sample followed by the CEM sample. Other two important parameters analyzed were the specific strength (SS) [49–51,54] and SS/$i_{corr}$. The latter represents the attained CS concatenated with their specific mass. This induces a strength associated with a lightweight effect. The former means the strength/lightweight of the concrete paste concatenated with corrosion behavior of the embedded steel bar. The highest values were those of the CC sample followed by the CEM sample. This induces that the CEM sample, which has lower cement content (~10 wt.%) than other two concretes examined, is a competitive mixture to be considered.

**Table 6.** The values of the consumption of cement (Co), the Co per compressive strength (CS), the specific mass (d), the determined specific strength (SS) per (d) and SS per icorr at 7 and 28 days for the CC, SAND and CEM samples examined.

| Mix | Co (kg/m$^3$) | Co/CS (*) (kg·m$^{-3}$·MPa$^{-1}$) | | d (kg/m$^3$) | SS (**) ($10^3 \times$ m$^2$/s$^2$) | SS/i$_{corr}$ (') ($10^5 \times$ N·m$^3$/A·kg) |
|---|---|---|---|---|---|---|
| CC | 368 | 7d | 20 ($\pm$3) | 2143 | 8.4 ($\pm$0.2) | 0.38 |
| | | 28d | **16 ($\pm$1)** | | **10.7 ($\pm$0.2)** | **0.55** |
| SAND | 370 | 7d | 26 ($\pm$3) | 2137 | 6.5 ($\pm$0.2) | 0.36 |
| | | 28d | 25 ($\pm$3) | | 7.0 ($\pm$0.1) | 0.30 |
| CEM | <u>**339**</u> | 7d | 23 ($\pm$1) | 2128 | 7.5 ($\pm$0.1) | 0.31 |
| | | 28d | **21 ($\pm$3)** | | **8.5 ($\pm$0.1)** | **0.40** |

(*) Co per CS represents the consumption of cement per compressive strength (CS); (**) SS means specific strength being CS per density (d, or specific mass) as a function of mixture elements; (') SS per i$_{corr}$ is ratio between SS and corrosion current density (i$_{corr}$) obtained from Table 2.

Additionally, it is worth noted that, at initial period, the CEM sample (containing fine EG particles) reveals that a passive layer is rapidly formed, when compared to other two examined concrete samples. This seems to imply a slight better corrosion behavior for a long-term immersion period.

Based on the aforementioned assertions, eggshell waste (independently of their origin; due to possess similar compositions [83,84]) or similar material (e.g., limestone [4–9]) is a promise material to be used in civil construction, replacing cement portion and a decrease of about 10% in cement consumption is reached. With this, incommensurable economical and environmentally friendly gains can be attained. On the one hand, it is recognized that the worldwide cement consumption is more than 4 billion tons (in 2020) [84] and eggshell can also be applied in other distinctive industrial applications [85,86]. On the other hand, it is believed that there exists feasible accountability to propose and adequate planning of the certain portion of the global chicken egg production (~80 million metric tons [86,87]) to be utilized in civil engineering (replacing structural concrete or cement paste applications since minimal compressive strength is attained).

**4. Conclusions**

From the experimental results, the following conclusions can be drawn:

- EIS technique and potentiodynamic polarization curves are useful to predict the corrosion behavior of the embedded steel bars. EIS technique permits to analyze double layer condition and concatenated planar and porous electrode behaviors were described.

- It is also found that ES particles in two distinct sizes provide different corrosion aspects. Coarser ES particles did not substantially affect the initial rebar passivation as observed in the CC (control/reference) sample. On the other hand, when finer ES particles were utilized, the resulting passive layer is formed at initial period of immersion. This seems to positively reflect on the corrosion behavior in a long-term immersion period. These conclusions are based on both EIS and polarization results.

- The eggshell content has no deleterious effect up on the fresh state properties. The resulting mechanical behavior is not substantially decreased. The concrete with 10 wt.% finer ES particles has their corresponding mechanical behavior decreased between ~10% and 12% when the CC (reference) mixture is compared.

- Concatenating both mechanical strength and lightweight effect, and associating with corrosion behavior, it is induced that the proposed mixture containing 10 wt.% finer eggshell (ES) particles, demonstrates interesting and competitive results. An incommensurable environmentally friendly aspect can also be associated.

- Since no substantial decreases in corrosion resistance is observed when eggshell is used, its replacement with cement portion (~10wt.%) represents significant economical gain in potential civil engineering application.

**Author Contributions:** Y.A.M. and I.M. developed the mixture and fresh and hardneded measurements and EIS/polarization analyses. R.S.B. and A.D.B. developed SEM observations and also helped with the mixture calculations and designing as well as with the general organization and English written; W.R.O. provided the general organization of the experiments and analyses of the correlation between microstructure and resulting properties. He has also written and organized the proposed manuscript. All authors have read and agreed to the published version of the manuscript.

**Funding:** Acknowledgments are also guided to the financial support provided by FAEPEX-UNICAMP (#2407/21 and # 2120/21), CAPES (Coordination for the Improvement of Higher Education Personnel, Ministry of Education, Brazil, Grant #1), CNPq (The Brazilian Research Council) Grants, #405602/2018-9 and #304950/2017-3.

**Institutional Review Board Statement:** Not applicable.

**Informed Consent Statement:** Not applicable.

**Data Availability Statement:** The authors also declare that all research data supporting this publication are directly available within this publication.

**Acknowledgments:** An anonymous colleague whom is a native speaker has significantly contributed with English writing revision. The authors acknowledge this mentioned colleague. Authors also acknowledge Luiz Antonio Garcia and Rosa C Ceche Lintz by important technical contributions.

**Conflicts of Interest:** The authors declare no conflict of interest.

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
