# Peer review of "EIS Investigation of the Corrosion Behavior of Steel Bars Embedded into Modified Concretes with Eggshell Contents"

_metals, doi:10.3390/met12030417_

Round 1

Reviewer 1 Report

The topic is surely worth to be investigated, but the paper is evidently much too long and consequently fairly complicated. Numerous issues are included, but the research work is poorly described and conclusions insufficiently supported. Mineralogy of different concretes is relatively well explained, but the relationship to corrosion it is not quite clear. Moreover, although EIS is contained in the title of the paper, the overall corrosion aspects are missing or hidden in the longer text.

Some of my specific comments are listed below:

  • References related to carbonates in concretes and EIS related to corrosion of steel in concrete are rich. Consequently, the introduction chapter is quite OK, but the main aims of the work are still unsure: change of mineralogy due to ES particles? change of EIS response due to ES? influence to corrosion processes?

  • There is poorly described how to specimens were exposed to chlorides (did I miss something?). The stationarity of corrosion processes during the first part of exposure (1th day, 7th day) was not considered (did I miss something?). The influence of potentiodynamic polarization curves to corrosion processes was not mentioned (EIS is considered to be “non-destructive”, what is not the case with potentiodynamic polarization curves; did I miss something?). The corrosion damage of steel in specimens were not evaluated, so the response of EIS could vary (Figure 8.).

  • It is not clear what was the origin/source of figures 7., 8. and 9: was this the interpretation of the research, or based on “general” knowledge/references. Label of Figure 4. is missing.

Author Response

Manuscript metals-1551977

Reviewer(s)' Comments to Author:

Reviewer: 1

The topic is surely worth to be investigated, but the paper is evidently much too long and consequently fairly complicated. Numerous issues are included, but the research work is poorly described and conclusions insufficiently supported. Mineralogy of different concretes is relatively well explained, but the relationship to corrosion it is not quite clear. Moreover, although EIS is contained in the title of the paper, the overall corrosion aspects are missing or hidden in the longer text.

AUTHORS: The Authors congratulate the comments provided. The main text was revised and improved. The weaknesses commented are revised and improved. Conclusions are also revised and better correlated with those results attained. Additionally, overall corrosion aspects are also revised and improved. These new corrections/improvements are included, as yellow highlighted text throughout the manuscript revised.

Some of my specific comments are listed below:

  • References related to carbonates in concretes and EIS related to corrosion of steel in concrete are rich. Consequently, the introduction chapter is quite OK, but the main aims of the work are still unsure: change of mineralogy due to ES particles? change of EIS response due to ES? influence to corrosion processes?

AUTHORS: The Reviewer is correct with this comment. In order to solve this main aim, a revision was meticulously provided and mentioned problem is solved. In the last paragraph of the revised manuscript, these corrections are yellow highlighted.

“The main aim of this proposed investigation concerns to evaluation of both mechanical and corrosion behavior when eggshell (ES) from chicken origin is used. With this, it is expected that the mineralogy of concrete is not substantially affected. However, the economical aspect is an interesting characteristic since no decreasing in corrosion is attained. Associated with this, novelty and contribution concern to the evaluation of corrosion behavior concatenated with mechanical properties. Besides, both porous and planar behaviors are responsible to predict the resulting corrosion behavior. In this investigation, the three distinct concrete mixtures are prepared, i.e. a reference (control) and two modified concretes.”

  • There is poorly described how to specimens were exposed to chlorides (did I miss something?). The stationarity of corrosion processes during the first part of exposure (1th day, 7thday) was not considered (did I miss something?).

AUTHORS: The Reviewer is also right. The mentioned weakness is revised and improved. Into the Experimental procedure section, the yellow highlighted texts are included, as follow:

“Before all experimentations, the samples were immersed in stagnant and naturally aerated 0.5M NaCl solution during 7 days. The samples were displaced in recipient containing NaCl with concrete bulk completely immersed (i.e. of about 100 mm considering the height of the cylindrical concrete sample. The top of the steel bar (i.e~ 30mm ) was not immersion in NaCl solution. After this period (stationary stage) in order to procedure both EIS and polarization tests, a volume of about 450 (±15) mL of a stagnant and naturally aerated 0.5 M NaCl at 25 (±2) oC and initial pH of about 6.5 (±0.5) is adopted.”

  • The influence of potentiodynamic polarization curves to corrosion processes was not mentioned (EIS is considered to be “non-destructive”, what is not the case with potentiodynamic polarization curves; did I miss something?). The corrosion damage of steel in specimens were not evaluated, so the response of EIS could vary (Figure 8.).

AUTHORS: Considering the comment provided, the sentences are included, as yellow highlighted.

“It is remarked that potentiodynamic polarization measurements have not substantially affected or damaged the steel bars (SB). Although it is recognized that EIS is a non-destructive testing, also no substantial damages on steel bars’ surfaces (covered by concrete bulk) are observed when polarization tests are carried out. Although the top SB (portion out of the bulk concrete) depicts red rust portion, this was cleaned (grounding) before each one of the experimentations. This is made to connect with potentiostat by using a jack connection, as shown in Fig. 1(b). It is also notice that after ~365 days no substantial damages are observed.”

  • It is not clear what was the origin/source of figures 7., 8. and 9: was this the interpretation of the research, or based on “general” knowledge/references. Label of Figure 4. is missing.

AUTHORS: In order to solve this confusion, the sentences corresponding with each one of the mentioned figures are included, as yellow highlighted.

CASE OF FIGURE 7:

Figure 7a,b,c show the schematic representations of Nyquist plots with planar, porous and porous with Warburg component (transport and diffusion), respectively. These Figures are proposed based on previous reported investigations [37-38, 56-57, 59] and associated with corrosion mechanism phenomena proposed to occur.

CASE OF FIGURE 8:

A schematic representation of the proposed EC considering concrete with distinctive mixture elements, interface (forming corrosion by-products) and steel bar is shown Figure 8. This is proposed based on previously reported circuits and adapting with proposed corrosion mechanism.

CASE OF FIGURE 9:

The representations for an initial immersion period of the three concretes are shown in Figure 9a,b,c. This is proposed based on phenomena typically occurring in corrosion systems and adapting with proposed mechanism for this investigation considering ES particles additions.

Reviewer 2 Report

This article aimed to assess the corrosion behavior of embedded steel bars (SB) into concretes modified with of eggshell contents. Electrochemical impedance spectroscopy (EIS) and potentiodynamic polarization techniques were used in this paper. The results of the study found that the porous electrode behavior could help predict the corrosion mechanism. In addition, finer eggshell (ES) particles in concrete mixes could quickly passivate the rebar to improve the corrosion behavior of the embedded rebar. The paper is interesting and useful for the practicing engineers. The following comments are suggested:

  1. In lines 265-267 of page 6, "Before initiate the electrochemical measurements, the concrete specimens are kept immersed in NaCl solution during 10 minutes with all electrodes connected. Before experimentations, it is considered that a quasi-steady state is attained." The authors are recommended to cite some references related to this.
  2. The authors are recommended to provide relevant specifications for EIS tests.
  3. Abbreviations should be presented in their original font on their first appearance.
  4. In lines 333-340 of page 9, the authors are recommended to cite some references related to this.
  5. Bode plot is a semi-logarithmic coordinate plot of the transfer function of a linear time-invariant system versus frequency. The frequency on the horizontal axis is expressed on a logarithmic scale, and the frequency response of the system can be seen by using the Bode plot. The Bode diagram is generally composed of two diagrams, one amplitude-frequency diagram represents the change of the decibel value of the frequency response gain against frequency, and the other phase-frequency diagram is the change of the phase of the frequency response against the frequency. The authors should give a brief description in the text.
  6. Nyquist plot is for a continuous-time linear time-invariant system, the gain and phase of its frequency response are plotted in the complex plane in polar coordinates, often used in control systems or signal processing, can be used to determine feedback whether the system is stable. Each point on the Nyquist plot corresponds to the frequency response at a specific frequency. The angle of the point relative to the origin represents the phase, and the distance from the origin represents the gain. Therefore, the Nyquist plot combines the amplitude and the Bode plots of the phases are combined in one graph. The authors should give a brief description in the text.
  7. Figure 4 on page 9 is missing the figure title.
  8. The author should explain the content of Figures 4d, e, f.
  9. In Figure 7, the figure number of each figure should be added.
  10. The authors should improve the resolution of Figure 4, Figure 5, Figure 11, Figure 13(g), and Figure 16.
  11. The author should further explain about ZImaginary in Figure 4 and Figure 5.
  12. In Figure 14, Figure 15, and Table 5, some garbled characters appear. Please correct.

Author Response

Reviewer(s)' Comments to Author:

Reviewer: 2

This article aimed to assess the corrosion behavior of embedded steel bars (SB) into concretes modified with of eggshell contents. Electrochemical impedance spectroscopy (EIS) and potentiodynamic polarization techniques were used in this paper. The results of the study found that the porous electrode behavior could help predict the corrosion mechanism. In addition, finer eggshell (ES) particles in concrete mixes could quickly passivate the rebar to improve the corrosion behavior of the embedded rebar. The paper is interesting and useful for the practicing engineers. The following comments are suggested:

  1. In lines 265-267 of page 6, "Before initiate the electrochemical measurements, the concrete specimens are kept immersed in NaCl solution during 10 minutes with all electrodes connected. Before experimentations, it is considered that a quasi-steady state is attained." The authors are recommended to cite some references related to this.

AUTHORS: The Reviewer is correct. The sentences are revised and suggestion adopted.

2. The authors are recommended to provide relevant specifications for EIS tests.

AUTHORS: It seems that Reviewer #1 has also same comments/suggestions provided. Based on this, more details concerning to EIS were included, as yellow highlighted.

3. Abbreviations should be presented in their original font on their first appearance.

AUTHORS: The Reviewer is correct. Throughout the main text, all abbreviations were revised and elucidated at this first appearance.

4. In lines 333-340 of page 9, the authors are recommended to cite some references related to this.

AUTHORS: The suggestion is adopted. Distinct block of references are included.

5. Bode plot is a semi-logarithmic coordinate plot of the transfer function of a linear time-invariant system versus frequency. The frequency on the horizontal axis is expressed on a logarithmic scale, and the frequency response of the system can be seen by using the Bode plot. The Bode diagram is generally composed of two diagrams, one amplitude-frequency diagram represents the change of the decibel value of the frequency response gain against frequency, and the other phase-frequency diagram is the change of the phase of the frequency response against the frequency. The authors should give a brief description in the text.

AUTHORS: The suggestion is adopted and yellow highlighted text included.

6. Nyquist plot is for a continuous-time linear time-invariant system, the gain and phase of its frequency response are plotted in the complex plane in polar coordinates, often used in control systems or signal processing, can be used to determine feedback whether the system is stable. Each point on the Nyquist plot corresponds to the frequency response at a specific frequency. The angle of the point relative to the origin represents the phase, and the distance from the origin represents the gain. Therefore, the Nyquist plot combines the amplitude and the Bode plots of the phases are combined in one graph. The authors should give a brief description in the text.

AUTHORS: Again, the suggestion is adopted. In this opportunity, the Authors congratulate the Reviewer for this comment/suggestion.

7. Figure 4 on page 9 is missing the figure title.

AUTHORS: Caption of Fig. 4 is revised and adequately replaced.

Figure 4. Experimental results of EIS of the CC, SAND and CEM samples at distinct curing/immersion curing periods (1, 7, 28 and 365 days) in a stagnant/naturally aerated 0.5 M NaCl solution. (a), (b) and (c) depicts Bode and Bode-phase plots and (d), (e) and (f) show Nyquist plots in different examined days.

8. The author should explain the content of Figures 4d, e, f.

AUTHORS: The Reviewer’s suggestion is adopted and new sentences are included, as yellow highlighted.

“Although Nyquist plots are forwardly discussed (In Fig. 7), it is remarked that Figure 4(d), (e) and (f) depict results in Nyquist representation of the examined sample (CC, SAND and CEM) after 1, 7, 28 and 365 days of immersion period. The doted red lines mean the straight lines at 45o with ZReal axis (component), which induces to porous electrode behavior, as will forwardly detailed. However, from this point, it can be said that the same trends observed when Bode and Bode-phase are also indicated.”

9. In Figure 7, the figure number of each figure should be added.

AUTHORS: The Reviewer is correct. The suggestion is adopted.

10. The authors should improve the resolution of Figure 4, Figure 5, Figure 11, Figure 13(g), and Figure 16.

AUTHORS: It was tried to provide this. However, it seems that resolution is intrinsically associated with website (peer reviewer system). A 600 dpi file will be attached in the last version of the each figure.

11. The author should further explain about ZImaginary in Figure 4 and Figure 5.

AUTHORS: The Reviewer suggestion is adopted. New sentences are included to elucidate this comment provided. These sentences are yellow highlighted.

12. In Figure 14, Figure 15, and Table 5, some garbled characters appear. Please correct.

AUTHORS: Authors apologize for this occurrence. The corrections were provided. This seems to be occurred when a PDF file is made/constructed.

Reviewer 3 Report

Dear Authors,
thank you for your paper focused on the corrosion of steel bars embedded into concretes modified with eggshell content. My comments are:
- line 43 - ... [3] and Nurul et al. [4] ...,
- line 70 - there is "yield stress" - what is "yield stress"? "Yield strength" is used for reinforcement (yield stress is an old term for yield strength, which is no longer used), but in the case of concrete, it is not used. In concrete we know "compression strength", or "tension strength" or "stress in concrete",
-p. 9, line 234 - codes ABNT BNR 15823 and ASTM C1611-18 - they are missing in references, please add them,
- line 243 and Equation (1) - "Ft" - "t" should be subscript, right?
- page 9 - missing the designation of Fig. 4,
- line 449, there is "Murray [71] ... - in references is Murry - Murry or Murray?
- fig. 11, very difficult to read the text in the pictures, small font,
- tab. 5 - some wrong description for icorr and ipp (under parameter),
- figs. 14-15 - what are the question mars?
- fig. 16 - increase the quality of the images on the right side, they are indistinct,
- general remark - this article is difficult to evaluate, on the one hand, it is an interesting scientific article at a relatively high scientific level, but I do not see the practical application of the results in practice. How do the authors imagine the use of eggshells in practice?
- What eggshells were used in the study? From chicken? And what about using other types of eggshells? Are they all the same composition?
- With such a large consumption of concrete for construction, this represents too many eggshells. Is it practically possible to obtain such a large number of eggshells?
- it would be appropriate to supplement the authors' opinion on practical use - where they see the use of this type of concrete, on what structures or elements.
Best regards.

Author Response

Manuscript metals-1551977

Reviewer(s)' Comments to Author:

Reviewer: 3

 Dear Authors, thank you for your paper focused on the corrosion of steel bars embedded into concretes modified with eggshell content. My comments are:

AUTHORS: Authors are very satisfied and happy with fine and precise revision provided by this Reviewer. Authors congratulate for fine comments and suggestions, which were incorporated at revised version of the proposed manuscript.

- line 43 - ... [3] and Nurul et al. [4] ...,

AUTHORS: Correction was adopted. Nurul was replaced with “Islam”.

- line 70 - there is "yield stress" - what is "yield stress"? "Yield strength" is used for reinforcement (yield stress is an old term for yield strength, which is no longer used), but in the case of concrete, it is not used. In concrete we know "compression strength", or "tension strength" or "stress in concrete",

AUTHORS: The Reviewer is correct. The sentence was modified.

-p. 9, line 234 - codes ABNT BNR 15823 and ASTM C1611-18 - they are missing in references, please add them, - line 243 and Equation (1) - "Ft" - "t" should be subscript, right? - page 9 - missing the designation of Fig. 4,

AUTHORS: The Reviewer is correct. The corrections were adopted.

- line 449, there is "Murray [71] ... - in references is Murry - Murry or Murray?

AUTHORS: The adequate correction was adopted.

- fig. 11, very difficult to read the text in the pictures, small font,

AUTHORS: Fig. 11 was reworked in order to solve the weakness commented.

- tab. 5 - some wrong description for icorr and ipp (under parameter),

AUTHORS: The Reviewer is correct. The corrections were provided.

- figs. 14-15 - what are the question mars?

AUTHORS: The Reviewer is correct. New sentences (yellow highlighted) are provided/reformulated.

- fig. 16 - increase the quality of the images on the right side, they are indistinct, - general remark - this article is difficult to evaluate, on the one hand, it is an interesting scientific article at a relatively high scientific level, but I do not see the practical application of the results in practice. How do the authors imagine the use of eggshells in practice?

AUTHORS: The Reviewer is correct. Based on this important comment, new sentences are included to elucidate the matter and comment proposed. These modifications are also yellow highlighted.

“Based on the aforementioned assertions, eggshell waste (independently of their origin; due to possess similar compositions [80]) or similar material (e.g. limestone [4-9]) is a promise material to be used in civil construction, replacing cement portion and a decrease of about 10% in cement consumption is reached. With this, incommensurable economical and environmentally friendly gains can be attained. On the one hand, it is recognized that the worldwide cement consumption is more than 4 billion tons (in 2020) [81] and eggshell can also be applied in other distinctive industrial applications [82]. On the other hand, it is believed that there exists feasible accountability to propose and adequate planning of the certain portion of the global chicken egg production (~80 million metric tons [83]) to be utilized in civil engineering (replacing structural concrete or cement paste applications since minimal compressive strength is attained).

- What eggshells were used in the study? From chicken? And what about using other types of eggshells? Are they all the same composition?

AUTHORS: This is another important comment. The chicken eggshell was used. However, due to very similar composition, other eggshell can also be replaced and used and similar results attained. New sentences and references are included, as demonstrated in previous item.

- With such a large consumption of concrete for construction, this represents too many eggshells. Is it practically possible to obtain such a large number of eggshells?

AUTHORS: Considering the Reviewer’s comment, new sentence and reference (at last paragraph of section 3) are also included in order elucidate and point out the mentioned question, as demonstrated in previous item.

- it would be appropriate to supplement the authors' opinion on practical use - where they see the use of this type of concrete, on what structures or elements. AUTHORS: Adopting the Reviewer’s suggestion, new sentences were included, as shown in previous items.

Round 2

Reviewer 3 Report

Dear Authors,

thank you for improving your paper and for incorporating comments. I have no further comments.

Best regards.

Author Response

The Authors  thanks you again.